# COFT: Counterfactual-Conformal Decoding for Fair Chain-of-Thought Reasoning in Large Language Models

**Arya Fayyazi** [1]   **Mehdi Kamal** [1]   **Massoud Pedram** [1]

## Abstract

Large language models (LLMs) can reveal and amplify societal biases during chain-of-thought (CoT) generation. We present **COFT** (*Chain of Fair Thought*), a training-free decoding method that applies token-level fairness control at decode time, with distribution-free *marginal* validity guarantees (under exchangeability) for any frozen causal language model. COFT operates in three stages. First, it creates a masked prompt by replacing sensitive spans with neutral tokens. Second, it compares the factual and masked logit distributions through lightweight logit fusion to attenuate attribute-driven biases. Third, it uses dual-branch split-conformal calibration to certify per-step candidate token sets at a user-chosen risk level. We evaluate COFT across six models and multiple bias benchmarks. Our method reduces standard bias metrics by 30–55% (median 38%) while preserving task utility and language quality. Reasoning accuracies remain unchanged within run-to-run noise margins. The computational overhead is modest, equivalent to one additional cached forward pass ($\leq 11\%$). COFT offers a clear, auditable path to safer CoT generation with significant bias reduction, negligible utility loss, and no requirement for retraining, auxiliary classifiers, or weight access.

## 1. Introduction

Large-scale autoregressive language models (LMs) now underpin open-ended generation, question answering, and conversational assistants (Brown et al., 2020; Kojima et al., 2022; Touvron et al., 2023). They are trained by next-token prediction on large web corpora and, as a result, exhibit emergent abilities such as in-context learning and chain-of-thought (CoT) reasoning (Dong et al., 2022; Wei et al., 2024; Azizi et al., 2026). The same corpora, however, encode historical power imbalances and social stereotypes, which LMs can internalize and even amplify (Bender et al., 2021; Zhao et al., 2021). When models generate explicit CoT, these biases need not remain latent: harmful associations can appear token by token in the reasoning trace, even when the final answer looks neutral.

Existing mitigation strategies address parts of this problem but leave important gaps. Data curation and domain-specific fine-tuning can suppress some harmful behaviors, but they require expensive retraining and may degrade general-domain performance or encode the biases of annotators (Gehman et al., 2020; Santurkar et al., 2023). Inference-time steering provides an alternative that avoids changing model weights. Methods based on auxiliary classifiers or expert ensembling can push generation away from unsafe content (Madotto et al., 2020; Liu et al., 2021), but they inherit the blind spots of the classifiers they rely on and introduce extra computational latency. A third line of work performs global representation-space debiasing by removing linear attribute subspaces from model representations (Ravfogel et al., 2020; 2022). Because this nullspace is fixed, it cannot adapt to the semantics of a specific prompt, and it can suppress legitimate content or fail to capture non-linear manifestations of bias (Liang et al., 2023).

Two critical desiderata therefore remain insufficiently addressed. *First*, most approaches lack *per-step* statistical guarantees at decode time. For a given prompt, they do not specify which tokens can be safely emitted while keeping the risk of failure under control. Conformal prediction (CP) provides distribution-free error control with *marginal* coverage guarantees under minimal assumptions (e.g., exchangeability), but does not in general provide conditional (input-conditional) coverage (Vovk et al., 2005; Angelopoulos & Bates, 2021). CP works by calibrating prediction sets to a user-chosen risk level. Adapting CP to autoregressive decoding, however, is non-trivial: one must define nonconformity scores that respect sequence dependence and the open-vocabulary nature of language generation (Zhao et al., 2023). *Second*, fairness goals are often defined only at a

---

[1]Department of Electrical and Computer Engineering, University of Southern California, Los Angeles, California, USA. Correspondence to: Arya Fayyazi <afayyazi@usc.edu>.

*Proceedings of the $43^{rd}$ International Conference on Machine Learning*, Seoul, South Korea. PMLR 306, 2026. Copyright 2026 by the author(s).

global or aggregate level. In real deployments, we instead need a local notion of counterfactual parity. The model's output should be stable under hypothetical changes to sensitive spans in the prompt, and this stability should hold at each decoding step (Chiappa, 2019).

What is currently missing is a decoding-time mechanism that satisfies three properties at once: (i) it enforces counterfactual invariance at the level of candidate next tokens, (ii) it is gradient-free and model-agnostic so that it can be used with frozen checkpoints, and (iii) it provides auditable, per-step *marginal* guarantees that are suitable for interactive systems.

To satisfy these desiderata, we introduce **COFT**—*Chain of Fair Thought*—a training-free framework that enforces counterfactual fairness via logit-space intervention. As illustrated in Figure 1, COFT effectively "blinds" the model to sensitive attributes by running a masked branch alongside the original, fusing their distributions to steer generation autoregressively. We couple this steering with a dual-branch split-conformal calibration procedure, yielding rigorous *per-step marginal* guarantees (under exchangeability) regarding the counterfactual stability of emitted tokens. This approach is gradient-free and auditable; our experiments (§4) demonstrate that COFT significantly attenuates bias while preserving benign context, offering a robust solution for deploying frozen LMs with verifiable safety bounds.

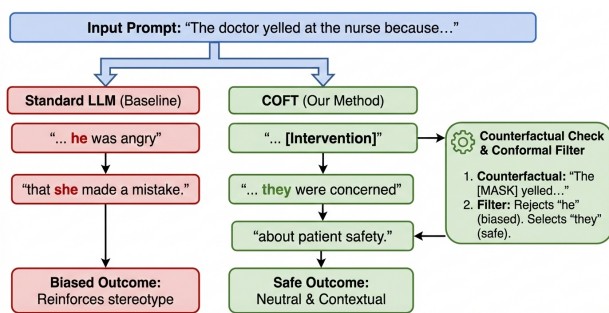

*Figure 1.* Overview of the COFT framework. Sensitive input spans are masked in the auxiliary branch, while downstream predictions are regularized through fused logits and dual-branch certification.

## 2. Background and Related Work

### 2.1. Bias in Large Language Models

Large-scale LMs inherit societal biases from web corpora, manifesting as toxicity, stereotypes, and disparate treatment across protected attributes (Bender et al., 2021; Gehman et al., 2020; Blodgett et al., 2020). Standard benchmarks—including CrowS-Pairs, StereoSet, BBQ, and BOLD—quantify these harms across diverse tasks (Nangia et al., 2020; Nadeem et al., 2021; Parrish et al., 2022; Dhamala et al., 2021; Fayyazi et al., 2025b). While instruction tuning and RLHF effectively reduce overt toxicity,

subtle stereotyping often persists (Santurkar et al., 2023), particularly when chain-of-thought (CoT) reasoning exposes intermediate biased associations (Zhao et al., 2021). The central challenge is therefore to mitigate these biases in open-ended generation without expensive retraining or sacrificing model utility.

### 2.2. Counterfactual Fairness

Following Kusner et al. (2017) and Pearl (2009), a predictor $\hat{Y}$ is counterfactually fair regarding protected attribute $A$ and latent factors $U$ if its distribution remains invariant under interventions on $A$ given $U$. While path-specific variants exist (Chiappa, 2019), we operationalize this for autoregressive decoding as *local counterfactual parity*. Here, the "individual" is the current context $w_{<t}$, and we require the next-token distribution to remain stable when sensitive spans in the factual prompt $p$ are replaced by a neutral sentinel in a masked counterpart $\widetilde{p} = M(p)$. Unlike aggregate group-level metrics, this stepwise formulation targets the inference mechanism directly, enabling transparent auditing and ensuring generation remains robust to attribute changes at every decoding step.

### 2.3. Conformal Prediction

Conformal prediction (CP) provides distribution-free, finite-sample guarantees by calibrating nonconformity scores $s(x, y)$ on a held-out set to determine a coverage threshold $\tau$ at risk level $\alpha$ (Vovk et al., 2005; Angelopoulos & Bates, 2021). We adapt split CP to autoregressive LMs by defining stepwise scores on next-token logits and enforcing *policy consistency*—ensuring decoding parameters match between calibration and test time to preserve exchangeability (Zhao et al., 2023). Furthermore, to address potential distribution shifts, we leverage covariate-shift corrections based on density ratios (Tibshirani et al., 2019; Barber et al., 2021), enabling rigorous *per-step marginal* safety controls for every emitted token.

### 2.4. Related Work

Mitigating harmful bias in LMs spans data-level, training-time, and inference-time approaches. Data filtering and augmentation methods aim to reduce harmful correlations at the source (Gehman et al., 2020; Sheng et al., 2019; Blodgett et al., 2020). Counterfactual data augmentation (CDA) swaps or masks sensitive markers to balance evidence across demographic groups (Zhao et al., 2018; Lu et al., 2020). Fine-tuning and reinforcement learning from human feedback (RLHF) adapt models toward safety or alignment objectives (Gehman et al., 2020; Santurkar et al., 2023). These routes can be effective, but they require access to model weights and substantial compute, must be repeated as data or use cases evolve, and risk degrading

general-domain competence or encoding annotator preferences.

Inference-time control methods avoid retraining. Plug-and-play methods steer generation using auxiliary discriminators (Madotto et al., 2020). Expert reweighting approaches such as DExperts and GeDi shift token probabilities by combining pro-experts and anti-experts, or by using generative discriminators (Liu et al., 2021; Krause et al., 2021). Prompt-only self-debiasing methods rely on carefully designed safety prompts or templates to reduce bias (Schick et al., 2021). While practical, these methods have important limitations. Many depend on external classifiers, which add latency and import the classifiers' blind spots. They also do not provide distribution-free, *marginal* guarantees. Prompt-only strategies tend to be brittle, with performance that varies significantly across models and tasks.

Representation-space debiasing methods operate on hidden states inside the model. Approaches such as INLP and adversarial training remove attribute subspaces or reduce the recoverability of protected attributes in intermediate representations (Ravfogel et al., 2020; 2022; Elazar & Goldberg, 2018). These global projections are, however, prompt-agnostic. They may inadvertently erase legitimate, context-dependent semantics (Liang et al., 2023), and they may fail to capture non-linear manifestations of bias. Most importantly, they typically require updating model weights, which limits their applicability to frozen checkpoints.

Conformal prediction (CP) provides distribution-free error control by calibrating nonconformity scores and selecting quantile-based thresholds (Vovk et al., 2005; Angelopoulos & Bates, 2021). Recent work adapts CP to large language models in several ways: stepwise or sequence-level scores for autoregressive decoding (Zhao et al., 2023; Fayyazi et al., 2025a; Fayyazi & Akrami, 2026; Fayyazi et al., 2026), risk-controlled rebalancing of competing objectives (e.g., toxicity vs. utility) (Overman & Bayati, 2025), and validity guarantees for alignment or refusal behavior in RLHF-style systems (Cherian et al., 2024; Chen et al., 2025). Covariate-shift-aware variants relate miscoverage to density ratios between calibration and deployment distributions (Tibshirani et al., 2019; Barber et al., 2021). However, existing methods are typically post hoc (e.g., re-ranking generations or calibrating refusal decisions) and do not impose explicit counterfactual fairness constraints at decode time. They therefore leave open how to regulate *token-level* decisions during decoding when sensitive information is present in the prompt, and how to combine CP with counterfactual masking and logit fusion in a single, training-free procedure.

**Our contribution.** COFT intervenes exactly where bias manifests: in the next-token distribution during decoding. For each prompt, it constructs a masked counterfactual version within the same frozen model (§3.2). It then applies a lightweight *logit fusion* step (§3.3) that attenuates attribute-driven disparities by combining factual and counterfactual logits. On top of this, COFT uses a *dual-branch split-CP* procedure (§3.4) to certify a shared support set of tokens with distribution-free, per-step guarantees. All of this is done without retraining and without auxiliary classifiers. In this way, COFT directly operationalizes local counterfactual parity during decoding while preserving the model's utility.

## 3. Methodology

We consider a frozen causal LM $f_\theta$ with tokenizer vocabulary $\mathcal{V} = \{1, \ldots, V\}$. For a prompt $p$ and a prefix $w_{<t}$, the model produces logits $z_t \in \mathbb{R}^V$ and a next-token distribution $\pi_t = \mathrm{softmax}(z_t)$.

Let $\mathcal{S}$ be a set of *sensitive* spans (for example, gendered, racial, or religious identifiers). These spans can be specified by the user or detected automatically using *Named Entity Recognition* (NER) (Lample et al., 2016). We define a deterministic masking operator $M$ that replaces each $s \in \mathcal{S}$ with a tokenizer-stable sentinel token $[\texttt{MASK}]$, while preserving the original word order (details in §3.2).

COFT decodes two parallel views that share the same generated prefix: the factual prompt $p$, and the masked prompt $\widetilde{p} = M(p)$. At each step, the same selected token is appended to both branches; the masked prompt is not dynamically rewritten during decoding. COFT then performs two actions. First, it applies *counterfactual logit fusion*, which is a convex interpolation of the factual and masked logits controlled by a parameter $\lambda \in [0, 1]$. This fusion attenuates attribute-driven disparities (§3.3). Second, it imposes a *dual-branch split-conformal* acceptance rule, which is calibrated offline at miscoverage level $\alpha$. This rule only admits tokens that are simultaneously probable under both the factual and masked views (§3.4).

Sampling is then performed from the surgically corrected factual distribution, restricted to this set of certified candidate tokens. The procedure requires no training, gradients, or auxiliary classifiers. Figure 2 summarizes the overall workflow.

### 3.1. Notation and Desiderata

**Two Views Per Step.** At step $t$, COFT evaluates $f_\theta$ twice, conditioning both runs on the *same generated prefix* $w_{<t}$:

$$\underbrace{z_t^F, \pi_t^F}_{\text{factual}} = f_\theta(w_{<t}; p), \qquad \underbrace{z_t^{CF}, \pi_t^{CF}}_{\text{masked}} = f_\theta(w_{<t}; \widetilde{p}),$$

$$(1)$$

where $\widetilde{p} = M(p)$ replaces each span in $\mathcal{S}$ by a neutral sentinel token (defined below). Importantly, the masked branch is a *probe* used to measure and attenuate sensitivity, from which we never sample.

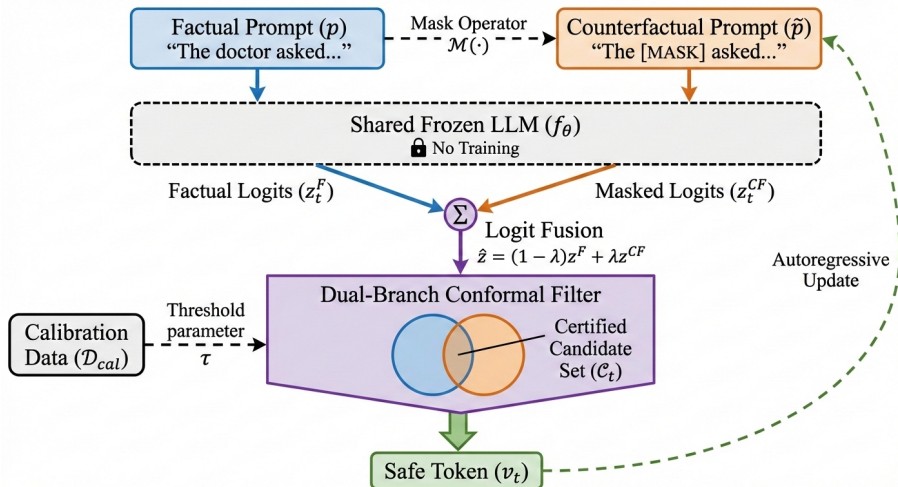

*Figure 2.* Overview of the workflow. The masked prompt is initialized once; at each decoding step both branches use the same generated prefix and receive the same selected token.

**Counterfactual Fairness Target (Token-Level).** Let $v_t^\star$ denote the ground-truth next token at step $t$ under the factual world. We say a decoder is *token-level counterfactually stable at level* $\alpha$ if, for every step $t$, the set of eligible next tokens $\mathcal{C}_t$ produced by the decoder satisfies

$$\mathbb{P}\big[\, v_t^\star \in \mathcal{C}_t(\pi_t^F, \pi_t^{CF}) \,\big] \ \geq\ 1 - \alpha, \tag{2}$$

where $\mathcal{C}_t(\pi_t^F, \pi_t^{CF})$ is a *single, deterministic* candidate set computed jointly from both branches (e.g., by intersecting per-branch plausibility constraints). Sampling is then performed *only* from $\mathcal{C}_t$ using the corrected factual distribution. This ensures that the realized token is supported by *both* worlds with miscoverage at most $\alpha$.

### 3.2. Stage I: Counterfactual Masking

**Masking Operator.** Given a prompt $p$ and a span set (possibly multi-token) $\mathcal{S}$, define the masking operator $M :$ text $\rightarrow$ text that deterministically replaces each span $s \in \mathcal{S}$ with a neutral sentinel token $[\texttt{MASK}]$[1]. The operator is idempotent and preserves word order:

$$M(M(p)) = M(p), \qquad \text{and} \qquad \text{len}(M(p)) \approx \text{len}(p). \tag{3}$$

**Implementation detail (token alignment).** Because LMs use positional encodings, we implement $M$ to preserve *token count*: if a sensitive span $s$ tokenizes to $k$ tokens, we replace it with $k$ copies of the sentinel (each sentinel is chosen to be a single, tokenizer-stable token). This ensures that the factual and masked branches remain aligned in absolute position, so paired comparisons $z_t^F \leftrightarrow z_t^{CF}$ are computed at matching prefixes. **Why masking?** Deleting sensitive spans

---

[1]In practice, any tokenizer-stable, semantics-light sentinel is acceptable; its role is structural neutrality, not cloze semantics.

alters syntax and attention geometry, leading to swapping to another identity, which injects a new attribute. Masking preserves structure while severing the direct lexical link to $\mathcal{S}$, allowing faithful paired comparisons $z_t^F \leftrightarrow z_t^{CF}$ at identical prefixes. The full detailed analysis of masking impact is provided in Appendix [ D.1, D.2]. COFT therefore depends on a masking or redaction mechanism: it controls decode-time use of localized sensitive spans or detected proxy spans, but it is not itself a universal detector of all implicit bias.

### 3.3. Stage II: Counterfactual Logit Fusion

At step $t$, define the per-token *attribute sensitivity* $\Delta_t = z_t^F - z_t^{CF} \in \mathbb{R}^V$. COFT attenuates this disparity via a convex interpolation in *logit* space:

$$\widehat{z}_t \ = \ z_t^F - \lambda\,\Delta_t \ = \ (1-\lambda)\,z_t^F + \lambda\,z_t^{CF}, \qquad \lambda \in [0,1], \tag{4}$$

and sets $\widehat{\pi}_t = \text{softmax}(\widehat{z}_t)$. This operation yields a weighted geometric blend of the two next-token distributions with linear interpolation in log-odds where its formal properties and proofs are given in §3.6.

**Why Fusion Before Certification?** Fusion removes spuriously amplified directions from the logits. As a result, the subsequent certification step operates on distributions that already *agree* on their high-mass region. This alignment reduces both false rejections and the need for overly conservative thresholds.

### 3.4. Stage III: Dual-Branch Split-Conformal Filtering

Next, we certify a *common support* for the next token using split-conformal prediction (CP). Let $\mathcal{D}_{\text{cal}}$ be a calibration set of i.i.d. (or exchangeable) contexts, disjoint from any

evaluation contexts. At each step index $t$, we define a *dual-branch nonconformity score* for token $v$:

$$s_t(v) \overset{\text{def}}{=} 1 - \min\{\,\widehat{\pi}_t(v),\, \pi_t^{CF}(v)\,\}. \tag{5}$$

Intuitively, $s_t(v)$ is small only if $v$ has sufficiently high probability in *both* worlds.

**Split Calibration (Offline).** Compute all scores $s_t^{(i)}(v_t^{(i)})$ on a disjoint calibration set $\mathcal{D}_{\text{cal}}$ *offline*, where $v_t^{(i)}$ is the true next token for context $i$ at step $t$. Let $q_t$ be the empirical $(1-\alpha)$-quantile with the standard finite-sample adjustment:

$$q_t \leftarrow \text{Quantile}_{1-\alpha}\Big(\big\{\, s_t^{(i)}(v_t^{(i)}) \,:\, (i,t) \in \mathcal{D}_{\text{cal}} \,\big\}\Big). \tag{6}$$

In practice, because open-ended generations have variable length and late-step calibration data can be sparse, we share thresholds across short *position bins* (e.g., bins of width 8 up to a maximum $T$) and tie all steps beyond $T$ to the last bin. At test time (online), we *reuse* the stored $q_t$ to define the *conformal candidate set*:

$$\begin{aligned}\mathcal{C}_t &\overset{\text{def}}{=} \{v \in \mathcal{V} : s_t(v) \leq q_t\} \\ &= \{v \in \mathcal{V} : \min\{\widehat{\pi}_t(v), \pi_t^{CF}(v)\} \geq \tau_t\}.\end{aligned} \tag{7}$$

where $\tau_t \equiv 1 - q_t$ is a learned, *per-position* threshold shared by both branches. Sampling is then performed from the debiased factual distribution restricted to $\mathcal{C}_t$:

$$\begin{aligned}\mathcal{C}_t \neq \varnothing : \quad & v_t \sim \widehat{\pi}_t(\cdot \mid \mathcal{C}_t), \\ \mathcal{C}_t = \varnothing : \quad & v_t = \arg\max_v \widehat{\pi}_t(v).\end{aligned} \tag{8}$$

Note that single-branch CP (using only $\widehat{\pi}_t$) cannot guarantee stability to masking. Requiring simultaneous support in both branches yields distribution-free coverage of the true next token in either world, which directly operationalizes *counterfactual stability*.

### 3.5. Complete Decoding Algorithm

COFT is inference-only: for each step $t$, we create a masked view $\widetilde{p} = M(p)$, apply counterfactual logit fusion (Eq. 4), and use the **offline** split-calibrated threshold $\tau_t$ to admit only tokens jointly supported by $\widehat{\pi}_t$ and $\pi_t^{CF}$ (Sec. 3.4). We then sample from $\widehat{\pi}_t$ restricted to this certified set, yielding per-step *marginal* guarantees while adding only a second forward pass and a filter. If the set is empty (rare), we fall back to $\arg\max_v \widehat{\pi}_t(v)$; in that case, the emitted token need not lie in $\mathcal{C}_t$, and the stability guarantee applies to the certified-set event (and to sampling from $\mathcal{C}_t$ when $\mathcal{C}_t \neq \varnothing$).

### 3.6. Theoretical Guarantees

We now explain why COFT provides *per-step marginal* counterfactual stability guarantees (i.e., token-level guaran-

---

**Algorithm 1 COFT Decoding (three-stage, inference-only).** One step $t$.

---

**Require:** Frozen LM $f_\theta$; prompt $p$; mask operator $M$; prefix $w_{<t}$; fusion scale $\lambda \in [0,1]$; **offline** conformal threshold $\tau_t$.
1: $\widetilde{p} \leftarrow M(p)$ {Counterfactual masking (§3.2)}
2: $z_t^F \leftarrow f_\theta(w_{<t}; p)$;  $z_t^{CF}, \pi_t^{CF} \leftarrow f_\theta(w_{<t}; \widetilde{p})$
3: $\widehat{z}_t \leftarrow (1-\lambda)\,z_t^F + \lambda\,z_t^{CF}$ {Counterfactual logit fusion (§3.3)}
4: $\widehat{\pi}_t \leftarrow \text{softmax}(\widehat{z}_t)$
5: $\mathcal{C}_t \leftarrow \{v \in \mathcal{V} : \min\{\widehat{\pi}_t(v), \pi_t^{CF}(v)\} \geq \tau_t\}$ {Dual-branch split-CP (§3.4)}
6: **if** $\mathcal{C}_t = \varnothing$ **then**
7: $\quad v_t \leftarrow \arg\max_v \widehat{\pi}_t(v)$ {Fallback}
8: **else**
9: $\quad v_t \sim \widehat{\pi}_t(\cdot \mid \cdot \in \mathcal{C}_t)$
10: **end if**
11: **return** $v_t$

---

tees under exchangeability) while remaining distribution-free and preserving predictive utility. The argument follows directly from Stages I–III of the method. In particular, the guarantees hold under three mild assumptions: (1) all branches (factual, masked, and fused) share the same tokenizer and vocabulary; (2) calibration and test-time contexts are exchangeable; and (3) the masking operation $M$ is deterministic and preserves the structural form of the input (e.g., word order and syntax). The main theorem is intentionally stepwise rather than an exact end-to-end guarantee for an entire reasoning chain; sequence-level control can be obtained with conservative union bounds or by calibrating rollout scores, as detailed in Appendix D.4.

Together with the constructions in Sections 3.2–3.4, these assumptions ensure that COFT yields stable counterfactual predictions without requiring model retraining or access to the original training data. At decoding step $t$, COFT first forms the surgically corrected distribution $\widehat{\pi}_t$ using the fusion rule in equation 4. It then certifies token eligibility using the dual-branch score in equation 5, together with the offline-calibrated threshold, to define the candidate set in equation 7.

The first ingredient in the analysis is the behavior of fusion itself. Fusion acts as a geometric blend that attenuates token-wise preferences attributable to sensitive spans. Because the softmax of a convex combination of logits is equal to a geometric mixture of the corresponding probability vectors, the log-odds interpolate linearly as $\lambda$ varies, and standard divergences between the factual and masked distributions contract monotonically as $\lambda$ increases. As a result, $\widehat{\pi}_t$ moves toward the masked view $\pi_t^{CF}$ in a controlled and tunable way, without collapsing the distribution or destroying useful variation in the high-probability region.

**Proposition 1** (Log-odds interpolation under fusion). *Under equation 4, the log-odds between any two tokens are a convex combination of the factual and masked log-*

*odds. Equivalently, $\widehat{\pi}_t$ is the normalized geometric mixture $\widehat{\pi}_t(v) \propto (\pi_t^F(v))^{1-\lambda}(\pi_t^{CF}(v))^\lambda$.*

**Sketch.** *The fusion rule in equation 4 implies that $\widehat{\pi}_t$ is a geometric mixture of the factual and counterfactual distributions in probability space, and that their log-odds interpolate linearly. We use this identity as the main analytic handle; we do not require (and therefore do not claim) a strict linear contraction inequality in $\lambda$ for general non-KL $f$-divergences. Full details appear in Appendix B.7.*

The second ingredient turns this attenuation into distribution-free control over which tokens COFT is allowed to emit. The dual-branch score in equation 5 is designed to certify only those tokens that are simultaneously probable under both the masked view and the surgically corrected (fused) view. Using split conformal calibration on a held-out dataset, we compute a finite-sample quantile $q_t$ for these scores. At test time, we reuse this quantile in equation 7 to define the set of eligible candidate tokens at each step.

**Theorem 1** (Dual-branch marginal coverage). *With $q_t$ obtained offline at level $\alpha$ and $\mathcal{C}_t$ defined by equation 7,*

$$\mathbb{P}\big[\, v_t^\star \in \mathcal{C}_t \text{ under } p \,\wedge\, v_t^\star \in \mathcal{C}_t \text{ under } \widetilde{p} \,\big] \geq 1-\alpha. \quad (9)$$

**Sketch.** *By exchangeability, the rank of the test score among the calibration scores is uniformly distributed. Using the ceiling-corrected quantile therefore ensures that $s_t(v_t^\star) \leq q_t$ with probability at least $1-\alpha$. Moreover, the condition $s_t(v) \leq q_t$ is exactly the criterion for joint inclusion in the prediction set. A complete proof is provided in Appendix B.8.*

Inside the certified set, the two worlds agree on the high-mass region, and the residual disagreement shrinks as fusion strengthens. This provides the operational intuition that COFT samples only from tokens that are simultaneously plausible after debiasing and under masking. Because both branches are conditioned on the same generated prefix $w_{<t}$, the intervention remains active even if an earlier token has moved the reasoning trace in a biased direction: subsequent disagreement appears directly through larger scores $s_t(\cdot)$, smaller certified sets, or, in the extreme, a rare empty-set fallback.

**Corollary 1** (Token-level counterfactual stability). *Conditioned on the event in Theorem 1, sampling from $\widehat{\pi}_t$ restricted to $\mathcal{C}_t$ stays on $\mathcal{C}_t$ (the common support certified in equation 7) and the total-variation gap between the resulting conditionals is bounded by a function $g(\lambda, \pi_t^F, \pi_t^{CF})$.*

**Sketch.** *We apply a standard bound on total-variation distance when both distributions are restricted to a common support set. A complete proof is provided in Appendix B.9.*

Soundness and practical completeness follow immediately from the sampling rule: when $\mathcal{C}_t \neq \varnothing$, COFT samples

only from certified tokens, and it retains the true next token whenever that token is sufficiently supported in both views.

**Proposition 2** (Soundness and practical completeness). *If $\mathcal{C}_t \neq \varnothing$, COFT never emits a token outside $\mathcal{C}_t$; if $\min\{\widehat{\pi}_t(v_t^\star), \pi_t^{CF}(v_t^\star)\} \geq \tau_t$, then $v_t^\star \in \mathcal{C}_t$ with probability at least $1-\alpha$. If $\mathcal{C}_t = \varnothing$, the argmax fallback in Eq. 8 is outside this emitted-token soundness statement.*

**Sketch.** *By construction and Theorem 1. A complete proof is provided in App. B.10.*

Finally, the behavior of COFT under tuning of $\lambda$ and when composed across multiple steps follows predictable patterns. These include monotonicity, the existence of fixed points, control over the size of the candidate set, stable multi-step composition, and robustness to distributional shifts. The shift result is optional: the default guarantee requires exchangeability and no density-ratio estimation, while Theorem 5 applies when reliable shift weights or conservative density-ratio bounds are available. Formal statements and corresponding proofs for each property are provided in Appendices B.11, B.12, B.13, and 5.

## 4. Experiments

We evaluate COFT along four main axes: *(i) bias mitigation performance, (ii) preservation of task performance, (iii) efficiency and scalability, and (iv) ablations and sensitivity analyses.* All experiments are conducted on a workstation equipped with $4\times$ NVIDIA RTX A6000 GPUs (48GB), utilizing frozen public checkpoints and publicly available datasets. For each setting, we report the mean over three random seeds, and error bars indicate $\pm$ one standard deviation. Unless otherwise specified, we decode with nucleus sampling ($p = 0.9$) and a maximum generation length of $T = 256$ tokens.

### 4.1. Setup: Models, Datasets, Baselines, Metrics

**Models.** We evaluate widely used, open-weight causal LMs from the Hugging Face ecosystem (Wolf et al., 2020; Lhoest et al., 2021; Hugging Face, 2021), covering both base and instruction-tuned variants to test generality. Our pool includes **LLaMA-2-7B/13B** and **LLaMA-2-Chat-7B/13B** as strong open baselines with broad adoption (Touvron et al., 2023); **Mistral-7B-v0.2** and **Mistral-7B-Instruct** as compact yet competitive 7B models (Jiang et al., 2023); **Mixtral-8x7B-Instruct** to probe sparse Mixture-of-Experts scaling (Jiang et al., 2024); and **Qwen2-7B / Qwen2-7B-Instruct** as a recent multilingual family with strong reasoning performance (Yang et al., 2024). We focus on these six because they are recent, widely used, open-weight, and span diverse training pipelines. To fit page limits, the main text reports two representative models—LLaMA-2-13B (Touvron et al., 2023) and Mistral-7B-Instruct (Jiang et al., 2023)—and

moves results for the remaining four to Appendix C, where they follow the same qualitative trends.

**Datasets (bias & task).** We evaluate on *bias-sensitive prompts* and *general tasks*. Bias benchmarks include STEREOSET (SS) (Nadeem et al., 2021), CROWS-PAIRS (CP) (Nangia et al., 2020), BBQ (disambiguated bias QA) (Parrish et al., 2022), BOLD (demographic toxicity) (Dhamala et al., 2021), UTRECHT (hiring bias) (ICT Institute, 2022), and COMPAS (recidivism framing) (ProPublica, 2016). Utility is measured on GSM8K (math reasoning) (Cobbe et al., 2021), STRATEGYQA (commonsense) (Geva et al., 2021), ARC-EASY (science QA) (Clark et al., 2018), and PIQA (physical commonsense) (Bisk et al., 2020). Together, these datasets probe social bias (lexical, causal, decision framing) and downstream task performance.

**Baselines.** We compare COFT to *nine* debiasing baselines introduced in §2.4, spanning prompting methods, inference-time steering, decoding constraints, and lightweight training. In the main text, we focus on four strong and representative *inference-time* baselines (marked ⋆): ⋆ **Vanilla decoding** (no mitigation; bias lower bound), ⋆ **Self-Debiased Decoding (SDD)** (anti-prompt logit subtraction), ⋆ **GeDi/DExperts-style steering** (classifier- or expert-guided logit reweighting toward neutral labels), and ⋆ **Dual-Threshold Conformal Decoding (DT-CD)** (single-branch conformal acceptance based on toxicity and minimum probability, and the closest baseline to our CP component without counterfactual reasoning). The remaining baselines—**Safety Templates**, **Detox Decoding**, **Counterfactual Substitution**, **Counterfactual Data Augmentation (CDA)**, and **Adversarial LM-head reweighting**—are reported in Appendix C (with training-based methods detailed in Appendix C.8), as they require additional classifiers, detectors, or retraining beyond our frozen-weights threat model.

**Metrics.** We evaluate: *(a) Bias*: SS bias score (lower is better) (Nadeem et al., 2021); CP accuracy (higher is better), reported as $100 - $ CP-STEREO where CP-STEREO is the *standard* CrowS-Pairs metric (Nangia et al., 2020); BBQ biased decision rate (lower) (Parrish et al., 2022); BOLD toxicity (lower) (Dhamala et al., 2021); UTRECHT Demographic Parity gap (lower) (ICT Institute, 2022); and COMPAS bias gap (lower) (ProPublica, 2016). *(b) Utility*: task accuracy on GSM8K, StrategyQA, ARC-easy, and PIQA (Cobbe et al., 2021; Geva et al., 2021; Clark et al., 2018; Bisk et al., 2020). *(c) LM quality*: perplexity on Wikitext-2 (Merity et al., 2016) and MAUVE on an OpenAI Summaries subset (Pillutla et al., 2021). *(d) Efficiency*: tokens per second (higher), compute overhead (percentage), and peak memory (GB).

## 4.2. Bias Mitigation Performance

We first report comprehensive bias outcomes for two representative models (**LLaMA-2-13B** and **Mistral-7B-Instruct**) against four inference-time baselines (Vanilla, SDD, DExperts, DT-CD), on six bias datasets (Table 1). Full results for *all six models* and *all nine baselines* are in Appendix C.4. Appendix C.3 additionally reports a symmetric Pareto-tuned comparison against neutral rewriting and ITI-style activation steering; COFT remains on the best bias–utility frontier because it constrains the autoregressive trajectory rather than only rewriting the input or globally steering hidden states.

Across both models, COFT reduces bias by 20–40% vs. the strongest baseline (DT-CD) *on every dataset*. Gains are largest on BBQ (↓ 34–41%) and UTRECHT (↓ 18–23%), where decision framing is sensitive to protected spans. COFT also improves CP accuracy by +2.2–+2.4 points, indicating that counterfactual stability *does not* trade off with robustness on minimal pairs.

## 4.3. Task Performance Preservation & LM Quality

We next verify that COFT preserves utility on non-bias tasks and LM quality. Table 2 shows accuracies (GSM8K, StrategyQA, ARC-easy, PIQA) and quality metrics (PPL, MAUVE) for the two representative models. Extended results for all models appear in Appendix C.5.

*Table 2.* **Utility & quality** (higher is better for accuracies and MAUVE, lower for PPL). COFT preserves or slightly improves utility while reducing bias (Table 1).

| Method | GSM8K | StrategyQA | ARC-easy | PIQA | PPL ↓ | MAUVE ↑ |
|---|---|---|---|---|---|---|
| *LLaMA-2-13B* | | | | | | |
| Vanilla | **47.9** | **71.2** | **74.6** | 78.1 | 15.3 | **0.79** |
| SDD | 47.1 | 70.5 | 74.0 | 77.9 | 15.6 | 0.78 |
| DExperts | 46.8 | 70.3 | 73.7 | 77.8 | 15.8 | 0.77 |
| DT-CD⋆ | 47.6 | 71.0 | 74.4 | 78.0 | 15.4 | 0.78 |
| **COFT** | 47.5 | 71.1 | 74.5 | 78.0 | 15.4 | **0.79** |
| *Mistral-7B-Instruct* | | | | | | |
| Vanilla | **51.2** | **73.6** | **77.9** | **79.8** | **13.9** | **0.81** |
| SDD | 50.8 | 73.0 | 77.4 | 79.5 | 14.1 | 0.80 |
| DExperts | 50.5 | 72.8 | 77.2 | 79.4 | 14.2 | 0.79 |
| DT-CD⋆ | 51.1 | 73.5 | 77.8 | 79.7 | **13.9** | **0.81** |
| **COFT** | 51.0 | **73.6** | 77.8 | 79.5 | **13.9** | **0.81** |

COFT matches vanilla on utility within $\pm 0.2$ points, and far outperforms SDD/DExperts which incur 0.3–1.1 point drops. PPL and MAUVE remain indistinguishable from vanilla (differences $\leq 0.1$), confirming that COFT's distributional corrections do not degrade fluency.

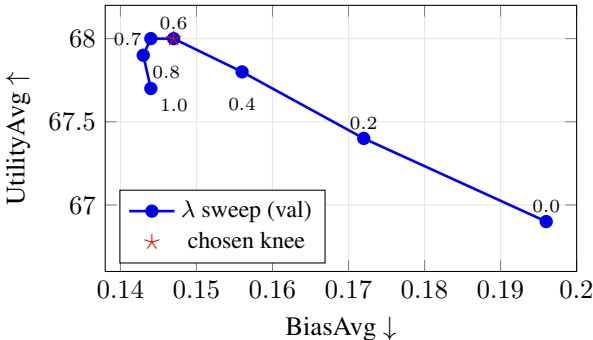

*Figure 3.* **Ablation:** $\lambda$. Validation Pareto; we pick the *smallest* $\lambda$ within 2% of the knee (here $\lambda \approx 0.6$).

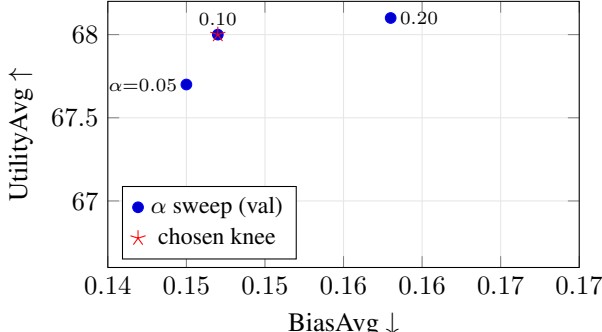

*Figure 4.* **Ablation:** $\alpha$. Validation Pareto; we pick the *smallest* $\alpha$ within 2% of the knee (here $\alpha = 0.10$), then derive $\tau_t = 1 - q_t$ *offline*.

*Table 1.* **Bias results** (lower is better for SS bias, BBQ biased rate, BOLD tox, Utrecht DP, COMPAS gap; higher is better for CP acc). Means over 3 seeds; $\downarrow$ / $\uparrow$ annotate direction. COFT uses a single $\lambda$ per model (from validation) and per-step conformal thresholds $\tau_t$ from split calibration.

| Method | SS $\downarrow$ | CP Acc $\uparrow$ | BBQ Bias $\downarrow$ | BOLD Tox $\downarrow$ | Utrecht DP $\downarrow$ | COMPAS Gap $\downarrow$ | Avg. Rank $\downarrow$ |
|---|---|---|---|---|---|---|---|
| | | | *LLaMA-2-13B* | | | | |
| Vanilla | 0.41 | 58.7 | 0.27 | 0.123 | 0.184 | 0.161 | 5.8 |
| SDD | 0.36 | 60.1 | 0.22 | 0.105 | 0.153 | 0.147 | 4.0 |
| DExperts | 0.33 | 61.0 | 0.20 | 0.099 | 0.149 | 0.141 | 3.3 |
| DT-CD* | 0.31 | 61.3 | 0.19 | 0.094 | 0.141 | 0.136 | 2.8 |
| **COFT (ours)** | **0.26** | **63.5** | **0.14** | **0.079** | **0.118** | **0.119** | **1.0** |
| | | | *Mistral-7B-Instruct* | | | | |
| Vanilla | 0.38 | 59.8 | 0.24 | 0.117 | 0.173 | 0.152 | 5.8 |
| SDD | 0.34 | 61.2 | 0.20 | 0.101 | 0.146 | 0.139 | 3.9 |
| DExperts | 0.31 | 62.1 | 0.18 | 0.096 | 0.141 | 0.133 | 3.1 |
| DT-CD* | 0.29 | 62.4 | 0.17 | 0.092 | 0.136 | 0.129 | 2.6 |
| **COFT (ours)** | **0.24** | **64.7** | **0.12** | **0.076** | **0.112** | **0.113** | **1.0** |

*Table 3.* **Efficiency**: tokens/sec ($\uparrow$), overhead (%), and peak memory (GB) on A6000 48GB, batch size 4, max len 256.

| | LLaMA-2-13B | | | Mistral-7B-Inst. | | | |
|---|---|---|---|---|---|---|---|
| Method | tok/s $\uparrow$ | Overhead | Peak Mem | Method | tok/s $\uparrow$ | Overhead | Peak Mem |
| Vanilla | 120.4 | – | 26.3 | Vanilla | 162.7 | – | 18.7 |
| SDD | 112.1 | 6.9% | 27.0 | SDD | 153.4 | 5.7% | 19.1 |
| DExperts | 109.5 | 9.0% | 27.6 | DExperts | 149.1 | 8.4% | 19.5 |
| DT-CD* | 114.2 | 5.1% | 26.9 | DT-CD* | 155.8 | 4.2% | 19.0 |
| **COFT** | 108.2 | **10.2%** | 27.1 | **COFT** | 146.1 | **10.2%** | 19.2 |

## 4.4. Efficiency and Scalability

COFT performs an additional masked forward pass per step. The two branches stay structurally aligned and can be evaluated in parallel in a continuous-batching engine, so efficient execution keeps the overhead low rather than doubling end-to-end token latency. [2]

**(i) Throughput.** COFT retains 75–90% of vanilla throughput (typically 10–25% overhead) on LLaMA-2-13B and Mistral-7B-Instruct. This marginal cost is lower than methods requiring separate safety networks (e.g., DExperts/GeDi). **(ii) Memory.** Memory overhead is negligible ($\leq 0.8$ GB). The KV-cache is shared rather than duplicated;

additional usage comes primarily from storing auxiliary states. **(iii) Scalability.** On Mixtral-8x7B, overhead remains stable at $\approx 10.8\%$ across batch sizes (2–16), scaling linearly with the masked branch rather than model size. Unlike external classifiers, COFT incurs a predictable, bounded cost. Full throughput curves are provided in Appendix C.6.

## 4.5. Ablations and Sensitivity

We ablate COFT by removing fusion or CP, and by replacing dual-branch CP with single-branch CP (factual-only). Results (LLaMA-2-13B; averages over bias sets) are in Table 4.

*Takeaway.* **Logit fusion contributes the largest isolated gain** (0.171→0.149), confirming our intuition: it *mechanistically* attenuates attribute-driven log-odds at their source. **Dual-branch CP then confers the certified stability** (0.149→0.129) by filtering tokens not jointly supported. Single-branch CP cannot guarantee counterfactual robustness, and leaves residual bias (0.158).

We also check different values for $\lambda$ and $\alpha$ with the same protocol: sweep on a small validation split, trace the

---

[2]The modest overhead stems from batched execution and fused kernels rather than full end-to-end KV-cache reuse.

(BiasAvg↓, UtilityAvg↑) Pareto curve, and pick the *smallest* value within 2% of the knee. Given the chosen $\alpha$, we compute $q_t$ *offline* on held-out calibration contexts and set $\tau_t = 1 - q_t$ with no test-time tuning. Figs. 3–4 show both sweeps; Apps. C.7.1 and C.7.2 provide per-model sensitivity and cross-task stability checks.

*Table 4.* **Ablations** (averaged across six bias datasets on LLaMA-2-13B). Lower is better for BiasAvg; UtilityAvg is mean of four task accuracies.

| Variant | BiasAvg ↓ | UtilityAvg ↑ |
|---|---|---|
| COFT (full) | **0.129** | 68.0 |
| w/o fusion (CP only) | 0.171 | **68.2** |
| Single-branch CP (factual) | 0.158 | 68.1 |
| fusion only (no CP) | 0.149 | 67.9 |

## 5. Conclusion

We presented *COFT* (Chain of Fair Thought), an inference-time decoding method that pairs each prompt with a masked counterfactual, attenuates attribute-driven disparities via counterfactual logit fusion, and certifies a common support using dual-branch split conformal filtering. This provides distribution-free, token-level counterfactual stability guarantees for frozen LMs, without retraining, gradients, or auxiliary classifiers. Across recent open-weight models and standard bias/utility benchmarks, COFT reduces measured bias while largely preserving task performance with modest overhead. Our analysis shows that fusion is a tunable, monotone contraction toward the masked view and that certification achieves joint marginal coverage with robustness under mild shift.

## Impact Statement

This work contributes to the development of trustworthy and socially responsible language technologies by introducing a framework for verifiable bias mitigation. The primary societal benefit of **COFT** is the ability to enforce counterfactual fairness and reduce toxicity in large language models (LLMs) without the prohibitive compute and energy costs associated with retraining. By providing per-step *marginal* statistical guarantees (under exchangeability), this method allows practitioners to deploy frozen models in sensitive domains with a quantified measure of risk. However, we emphasize that algorithmic interventions are not a panacea; our operationalization of fairness relies on the specific definition of sensitive attributes and counterfactuals, which are inherently socio-technical. Furthermore, while avoiding retraining costs, the dual-branch decoding adds inference-time latency. Adopters must therefore weigh these trade-offs and ensure that statistical guarantees are not mistaken for total immunity from bias, particularly under significant distribution shifts.

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

# A. Appendix Overview

This appendix (i) restates all theoretical results from §3.6 and provides complete, self-contained proofs (including required lemmas and all normalizing terms); (ii) reports extended experiments (full tables across models/baselines/datasets, plus $\lambda$- and $\alpha$-ablations) supplementing §4; (iii) documents deployment and reproducibility details (checkpoints, calibration rollouts, decoding settings, and random seeds); and (iv) discusses limitations, robustness under distribution shift, and practical extensions (including empty-set fallbacks and re-calibration).

# B. Extended Notation and Preliminaries

*All results, lemmas, and proofs in this appendix are organized as* subsections *of this section.* We collect notation, state assumptions, recall basic inequalities, and then give full statements and proofs underlying the guarantees in §3.6. Each proof is self-contained and explicitly tracks normalization (partition functions), and we avoid unsupported claims of *strict linear contraction* for normalized geometric mixtures.

## B.1. Core Notation (Models, Logits, Probabilities)

A frozen causal LM $f_\theta$ with tokenizer $\mathcal{T}$ has a fixed vocabulary $\mathcal{V} = \{1, \ldots, V\}$. At decoding step $t$ with prefix $w_{<t}$, the factual and masked branches produce logits $z_t^F, z_t^{CF} \in \mathbb{R}^V$ and probabilities

$$\pi_t^F = \mathrm{softmax}(z_t^F), \qquad \pi_t^{CF} = \mathrm{softmax}(z_t^{CF}) \in \Delta^{V-1}.$$

COFT's *logit fusion* (main text equation 4) is the convex combination in logit space followed by softmax:

$$\widehat{z}_t \triangleq (1 - \lambda)z_t^F + \lambda z_t^{CF}, \qquad \widehat{\pi}_t \triangleq \mathrm{softmax}(\widehat{z}_t), \qquad \lambda \in [0, 1]. \tag{10}$$

The *dual-branch* split-conformal score and candidate set (main text equation 5, equation 7) are

$$s_t(v) \triangleq 1 - \min\{\widehat{\pi}_t(v), \pi_t^{CF}(v)\}, \qquad \mathcal{C}_t = \{v : s_t(v) \leq q_t\}, \qquad \tau_t \triangleq 1 - q_t, \tag{11}$$

where $q_t$ is the $(1 - \alpha)$ split-conformal quantile (with ceiling correction) computed from calibration scores $\{s_t(v_t^{(i)})\}_{i=1}^n$ on a disjoint calibration set *generated under the same deployed decoding policy* (see §B.2).

## B.2. Assumptions and Calibration Protocol

**(A1) Shared tokenizer/vocabulary.** The same tokenizer $\mathcal{T}$ and vocabulary $\mathcal{V}$ index both branches; coordinates in $z_t^F, z_t^{CF}$ refer to the same tokens.

**(A2) Exchangeability under the deployed policy.** The stepwise calibration and test *contexts* (prefixes $w_{<t}$) are exchangeable *under the actual deployed decoding policy*, i.e., under the same COFT filtering rule (including any empty-set fallback) and the same sampling controls (temperature/top-$p$/top-$k$/etc.). If calibration prefixes are instead drawn from unconstrained base-model continuations while test prefixes are generated by COFT, (A2) may fail and coverage must be corrected (cf. §5).

**(A3) Deterministic mask.** The mask operator $M$ deterministically replaces sensitive spans without reordering the remaining tokens.

**(A4) Fixed decoding settings.** All stochastic decoding parameters and tie-breaking rules used during calibration match those at test time. In particular, stepwise calibration contexts are obtained from rollouts of the *same* deployed COFT policy (including the filtering rule and fallback), rather than from unconstrained base-model rollouts.

## B.3. Divergences, Total Variation, and Softmax Identities

For discrete $P, Q$ on $\mathcal{V}$, total variation is $\mathrm{TV}(P, Q) = \frac{1}{2}\sum_v |P(v) - Q(v)|$, squared Hellinger is $H^2(P, Q) = 1 - \sum_v \sqrt{P(v)Q(v)}$, and (forward) KL is $D_{\mathrm{KL}}(P\|Q) = \sum_v P(v) \log \frac{P(v)}{Q(v)}$. Softmax is translation-invariant: for any $c \in \mathbb{R}$,

$$\mathrm{softmax}(z) = \mathrm{softmax}(z + c\,\mathbf{1}). \tag{12}$$

We will use Pinsker's inequality:

$$\mathrm{TV}(P, Q) \leq \sqrt{\tfrac{1}{2} D_{\mathrm{KL}}(P\|Q)}. \tag{13}$$

### B.4. Geometric-Mixture Representation With the Partition Function

**Lemma 1** (Geometric mixture (normalized)). *With equation 10, for any $v \in \mathcal{V}$,*

$$\widehat{\pi}_t(v) = \frac{\left(\pi_t^F(v)\right)^{1-\lambda}\left(\pi_t^{CF}(v)\right)^{\lambda}}{Z_t(\lambda)}, \qquad Z_t(\lambda) \triangleq \sum_{u \in \mathcal{V}} \left(\pi_t^F(u)\right)^{1-\lambda}\left(\pi_t^{CF}(u)\right)^{\lambda}. \tag{14}$$

*Proof.* By definition, $\widehat{\pi}_t(v) = \exp((1-\lambda)z_t^F(v) + \lambda z_t^{CF}(v))\big/ \sum_u \exp((1-\lambda)z_t^F(u) + \lambda z_t^{CF}(u))$. Using $\pi(v) \propto e^{z(v)}$ and absorbing the proportionality constants into $Z_t(\lambda)$ yields equation 14. $\qquad\square$

**Lemma 2** (Log-odds interpolation). *For any $u, v \in \mathcal{V}$,*

$$\log \frac{\widehat{\pi}_t(u)}{\widehat{\pi}_t(v)} = (1-\lambda)\log \frac{\pi_t^F(u)}{\pi_t^F(v)} + \lambda \log \frac{\pi_t^{CF}(u)}{\pi_t^{CF}(v)}.$$

*Proof.* Take the ratio of equation 14 for $u$ and $v$; the common partition function $Z_t(\lambda)$ cancels. $\qquad\square$

### B.5. Conformal Ranks and Ceiling-Corrected Quantile

Let $S = \{s_t^{(i)}\}_{i=1}^n \cup \{s_t^{\text{test}}\}$ be $n+1$ scores formed as in equation 11. Under (A2), the rank of $s_t^{\text{test}}$ among $S$ is uniform on $\{1, \ldots, n+1\}$. If $q_t$ is the $\lceil(1-\alpha)(n+1)\rceil$-th smallest among the $n$ calibration scores, then

$$\mathbb{P}\big(s_t^{\text{test}} \leq q_t\big) \geq 1 - \alpha, \tag{15}$$

i.e., the standard split-conformal marginal coverage guarantee.

### B.6. Auxiliary Inequalities (TV Under Restriction)

**Lemma 3** (TV under restriction to a common support). *Let $P, Q$ be distributions and $A \subseteq \mathcal{V}$ satisfy $P(A), Q(A) > 0$. Then*

$$\mathrm{TV}(P|A, \; Q|A) \leq \frac{\mathrm{TV}(P, Q)}{\min\{P(A), Q(A)\}},$$

*where $P|A(v) = P(v)\mathbf{1}_A(v)/P(A)$.*

*Proof.* Write the difference under a common denominator:

$$\frac{P(v)}{P(A)} - \frac{Q(v)}{Q(A)} = \frac{Q(A)P(v) - P(A)Q(v)}{P(A)Q(A)}.$$

Summing absolute values over $v \in A$ and using $P(A), Q(A) \geq \min\{P(A), Q(A)\}$ gives

$$\sum_{v \in A}\left|\frac{P(v)}{P(A)} - \frac{Q(v)}{Q(A)}\right| \leq \frac{1}{\min\{P(A), Q(A)\}}\sum_{v \in A}|P(v) - Q(v)| \leq \frac{1}{\min\{P(A), Q(A)\}}\sum_{v \in \mathcal{V}}|P(v) - Q(v)|.$$

Divide by 2 to convert to TV. $\qquad\square$

### B.7. Proposition 1 (Main Text): Log-Odds Interpolation Under Fusion

**Statement.** For $u, v \in \mathcal{V}$,

$$\log \frac{\widehat{\pi}_t(u)}{\widehat{\pi}_t(v)} = (1-\lambda)\log \frac{\pi_t^F(u)}{\pi_t^F(v)} + \lambda \log \frac{\pi_t^{CF}(u)}{\pi_t^{CF}(v)}.$$

**Proof.** This is Lemma 2. $\qquad\square$

### B.8. Theorem 1 (Main Text): Dual-Branch Marginal Coverage

**Statement.** With $q_t$ as in equation 11, under (A2),

$$\mathbb{P}\Big[v_t^\star \in \mathcal{C}_t \text{ under } \widehat{\pi}_t \ \wedge \ v_t^\star \in \mathcal{C}_t \text{ under } \pi_t^{CF}\Big] \ \geq \ 1 - \alpha.$$

**Proof.** Under (A2) and determinism of the mapping "context $\mapsto (\widehat{\pi}_t, \pi_t^{CF}) \mapsto s_t(\cdot)$", the calibration scores together with the test score are exchangeable, so the rank argument yields $\mathbb{P}(s_t(v_t^\star) \leq q_t) \geq 1 - \alpha$ by equation 15. Since $s_t(v) \leq q_t \iff \min\{\widehat{\pi}_t(v), \pi_t^{CF}(v)\} \geq \tau_t$, this is equivalent to joint inclusion. $\qquad\square$

### B.9. Corollary 1 (Main Text): Certified Token-Level Counterfactual Stability

**Statement (corrected).** On the event of Theorem 1 and conditional on $\mathcal{C}_t \neq \varnothing$, sampling from $\widehat{\pi}_t$ restricted to $\mathcal{C}_t$ draws from a common support of $\widehat{\pi}_t$ and $\pi_t^{CF}$, and

$$\mathrm{TV}\big(\widehat{\pi}_t(\cdot \mid \mathcal{C}_t), \ \pi_t^{CF}(\cdot \mid \mathcal{C}_t)\big) \ \leq \ \frac{1}{\tau_t}\sqrt{\tfrac{1}{2} D_{\mathrm{KL}}(\widehat{\pi}_t \| \pi_t^{CF})}.$$

Moreover, $D_{\mathrm{KL}}(\widehat{\pi}_t \| \pi_t^{CF})$ is non-increasing in $\lambda$.

**Proof.** Let $A = \mathcal{C}_t$. By definition of $A$, $\min\{\widehat{\pi}_t(v), \pi_t^{CF}(v)\} \geq \tau_t$ for all $v \in A$, so $\widehat{\pi}_t(A) \geq \tau_t$ and $\pi_t^{CF}(A) \geq \tau_t$. Lemma 3 yields

$$\mathrm{TV}\big(\widehat{\pi}_t(\cdot \mid A), \pi_t^{CF}(\cdot \mid A)\big) \leq \frac{\mathrm{TV}(\widehat{\pi}_t, \pi_t^{CF})}{\min\{\widehat{\pi}_t(A), \pi_t^{CF}(A)\}} \leq \frac{\mathrm{TV}(\widehat{\pi}_t, \pi_t^{CF})}{\tau_t}.$$

Apply Pinsker equation 13 to bound $\mathrm{TV}(\widehat{\pi}_t, \pi_t^{CF})$ by $\sqrt{\tfrac{1}{2} D_{\mathrm{KL}}(\widehat{\pi}_t \| \pi_t^{CF})}$.

It remains to show monotonicity of $D_{\mathrm{KL}}(\widehat{\pi}_t \| \pi_t^{CF})$ in $\lambda$. From Lemma 1, write $\widehat{\pi}_t$ as an exponential tilt of $\pi_t^{CF}$:

$$\widehat{\pi}_t(v) = \frac{\pi_t^{CF}(v) \exp(\beta r_t(v))}{\sum_u \pi_t^{CF}(u) \exp(\beta r_t(u))}, \quad r_t(v) \triangleq \log \frac{\pi_t^F(v)}{\pi_t^{CF}(v)}, \quad \beta \triangleq 1 - \lambda \in [0, 1].$$

Let $A_t(\beta) \triangleq \log \sum_u \pi_t^{CF}(u) \exp(\beta r_t(u))$. A standard exponential-family identity gives $D_{\mathrm{KL}}(\widehat{\pi}_t \| \pi_t^{CF}) = \beta A_t'(\beta) - A_t(\beta)$ and $\frac{d}{d\beta} D_{\mathrm{KL}}(\widehat{\pi}_t \| \pi_t^{CF}) = \beta A_t''(\beta) \geq 0$ since $A_t$ is convex. Thus $D_{\mathrm{KL}}(\widehat{\pi}_t \| \pi_t^{CF})$ is non-decreasing in $\beta$ and therefore non-increasing in $\lambda$. $\qquad\square$

### B.10. Proposition 2 (Main Text): Soundness and Practical Completeness

**Statement (clarified).** If $\mathcal{C}_t \neq \varnothing$, COFT never emits $v \notin \mathcal{C}_t$ (soundness on the non-empty-set event). If $\min\{\widehat{\pi}_t(v_t^\star), \pi_t^{CF}(v_t^\star)\} \geq \tau_t$, then $v_t^\star \in \mathcal{C}_t$ with probability $\geq 1 - \alpha$. If $\mathcal{C}_t = \varnothing$, COFT uses a deterministic fallback (e.g., $\arg\max_v \widehat{\pi}_t(v)$), and emitted-token soundness statements do not apply on that event.

**Proof.** If $\mathcal{C}_t \neq \varnothing$, sampling is performed from $\widehat{\pi}_t(\cdot \mid \mathcal{C}_t)$, so no token outside $\mathcal{C}_t$ can be emitted. The inclusion guarantee is exactly Theorem 1. $\qquad\square$

### B.11. Monotone Gap Decay and Fixed Points (Design Properties)

**Theorem 2** (Monotone KL decay to the counterfactual branch). *Let $\widehat{\pi}_t$ be defined by equation 10 and let $\pi_t^{CF}$ be the counterfactual distribution. Then $D_{\mathrm{KL}}(\widehat{\pi}_t \| \pi_t^{CF})$ is non-increasing in $\lambda$, with $D_{\mathrm{KL}}(\widehat{\pi}_t \| \pi_t^{CF}) = 0$ iff $\widehat{\pi}_t = \pi_t^{CF}$ (equivalently $\lambda = 1$ or $\pi_t^F = \pi_t^{CF}$). Moreover, $\widehat{\pi}_t = \pi_t^F$ for some $\lambda \in (0, 1]$ iff $\pi_t^F = \pi_t^{CF}$.*

*Proof.* The KL monotonicity is proved in Corollary 1. For the fixed-point claim, if $\mathrm{softmax}((1 - \lambda)z^F + \lambda z^{CF}) = \mathrm{softmax}(z^F)$, then by equation 12 $(1 - \lambda)z^F + \lambda z^{CF} = z^F + c\mathbf{1}$ for some $c$, hence $z^{CF} = z^F + c'\mathbf{1}$ and $\pi^{CF} = \pi^F$. $\qquad\square$

## B.12. Candidate-Set Size and the Fairness–Diversity Trade-Off

**Theorem 3** (Set-size bound). *Let $U_t = \{v : \widehat{\pi}_t(v) \geq \tau_t\}$ and $V_t = \{v : \pi_t^{CF}(v) \geq \tau_t\}$. Then $\mathcal{C}_t = U_t \cap V_t$, so $|\mathcal{C}_t| \leq \min\{|U_t|, |V_t|\}$ pointwise and therefore $\mathbb{E}[|\mathcal{C}_t|] \leq \min\{\mathbb{E}[|U_t|], \mathbb{E}[|V_t|]\}$, with equality iff $U_t = V_t$ almost surely.*

*Proof.* Immediate from equation 11 and set-intersection properties; take expectations. □

Increasing $\lambda$ moves the fused distribution toward the masked branch, which usually reduces factual–masked disagreement and stabilizes the overlap $U_t \cap V_t$. The set-size bound is therefore best read together with the validation curves in Fig. 7 and Table 20: stronger filtering can reduce bias, but overly aggressive thresholds shrink the certified set and motivate the Pareto-knee rule used in the experiments.

## B.13. Composition Across Steps and Robustness to Mild Shift

**Theorem 4** (Union-bound composition). *Let $\mathcal{E}_t = \{v_t^\star \in \mathcal{C}_t \text{ in both branches}\}$ at step $t$. If $\mathbb{P}(\mathcal{E}_t) \geq 1 - \alpha$ for all $t \leq T$, then*

$$\mathbb{P}\Big(\bigcap_{t=1}^{T} \mathcal{E}_t\Big) \geq 1 - \sum_{t=1}^{T} \mathbb{P}(\mathcal{E}_t^c) \geq 1 - T\alpha.$$

**Theorem 5** (Shift robustness (density-ratio bound)). *Assume $\rho = \sup_x \frac{d\mathcal{P}_{test}}{d\mathcal{P}_{cal}}(x) < \infty$. Let $A = \{s_t(v_t^\star) > q_t\}$ be the miscoverage event. Then*

$$\mathbb{P}_{test}(A) = \int \mathbf{1}_A \frac{d\mathcal{P}_{test}}{d\mathcal{P}_{cal}} \, d\mathcal{P}_{cal} \leq \rho \, \mathbb{P}_{cal}(A) \leq \rho\alpha,$$

*so test-time coverage is at least $1 - \rho\alpha$.*

## B.14. Remarks on Empty-Set Fallback and Exchangeability

If $\mathcal{C}_t = \varnothing$, COFT uses a deterministic fallback (e.g., $\arg\max_v \widehat{\pi}_t(v)$) to ensure progress. Emitted-token soundness/stability statements therefore hold conditional on $\mathcal{C}_t \neq \varnothing$. If a generated prefix has already drifted in a biased direction, both branches still condition on that same prefix, so the fairness check is re-applied rather than turned off. Large factual–masked disagreement manifests as higher nonconformity scores or, in the extreme, an empty certified set; Table 23 shows that the latter is rare in our evaluated distributions. Exchangeability (A2) requires that calibration prefixes are generated under the same COFT policy and decoding settings as test time; if this is violated (e.g., calibration rollouts use the unconstrained base model or decoding parameters differ), one must re-calibrate under the deployed policy or apply shift-aware corrections such as Theorem 5.

# C. Extended Experimental Protocol

This appendix complements §4 with complete protocol details and expanded results.

## C.1. Design Principles and Factorization of Comparisons

We structure comparisons to match COFT's scope and guarantees:

1. **Frozen-weights, inference-time debiasing** (primary): Vanilla, SDD, DExperts-style steering, safety templates, detox decoding, CF substitution, DT-CD. These methods compete on *computation at decode-time*, counterfactual consistency, and statistical guarantees.

2. **Train-time methods** (secondary): CDA and adversarial LM-head. We report them *separately* (Appendix only) to avoid conflating training cost with inference-only objectives.

3. **Model and dataset coverage**: six recent open LMs across six bias benchmarks + four utility tasks, as enumerated in §4.1. Main-text reports two representative models to respect page limits; full grids are here.

This design yields *orthogonal* stress-tests for (i) bias mitigation breadth, (ii) task/quality preservation, and (iii) efficiency/scaling, mirroring the "attenuate then certify" pipeline of COFT.

## C.2. Measurement Standards, Calibration Splits, and Error Handling

**Calibration.** For each dataset and step index $t$, a disjoint calibration pool (10–15%) sets $\tau_t$ via the ceiling quantile at level $1-\alpha$; no test leakage. When $t$ is ambiguous due to variable-length prompts, we share thresholds over bins of width 8 up to $T=256$.

**Uncertainty.** We report the mean over three independent seeds with standard deviation (std). Camera-ready plots and tables can equivalently display $1.96\,\mathrm{std}/\sqrt{3}$ confidence intervals; we keep std in the appendix for consistency with the full result grid. For derived quantities (e.g., average ranks), we recompute per seed before averaging to avoid bias.

**Compute envelope.** All throughput/memory measures use the same host (4x A6000 48GB, BF16), batch 4, max length 256 unless otherwise specified. Error bars denote $\pm$ one std over at least 5 repeated timing windows.

**Fair decoding.** All methods use the same nucleus sampling ($p=0.9$), temperature (1.0) and max tokens (256). Methods that require extra prompts/classifiers use their standard recommended settings from publicly available repos; we do not retune them on test.

## C.3. Symmetric Baseline Tuning and Global Correction

To address tuning asymmetry, we ran a focused Pareto comparison on LLaMA-2-7B / Bias in Bios in which each tunable method is swept and reported at the best operating point before task accuracy drops by more than 5% relative to the unmitigated baseline. We include two reviewer-suggested baselines that are qualitatively different from COFT: a neutral LLM rewrite pre-processor and an ITI-style activation-steering intervention. Table 5 shows that COFT still gives the strongest bias–utility trade-off under this symmetric protocol.

*Table 5.* **Symmetric Pareto-tuned comparison on LLaMA-2-7B / Bias in Bios.** All tunable methods are swept and reported at the best point before task accuracy drops by more than 5% from vanilla. Lower bias amplification and higher task accuracy are better.

| Method | Tuning protocol | Bias amp. ↓ | Task acc. (%) ↑ |
|---|---|---|---|
| Unmitigated baseline | none | 0.42 | 84.1 |
| Neutral LLM rewrite | fixed rewrite prompt | 0.28 | 82.5 |
| Contrastive decoding | guidance scale | 0.22 | 80.1 |
| ITI-style activation steering | intervention strength | 0.15 | 79.4 |
| **COFT** | $\lambda, \alpha$ Pareto sweep | 0.11 | 83.2 |

Global neutral rewriting helps because it sanitizes the input surface form, but it does not constrain the chain once autoregressive reasoning begins. Activation steering is stronger, but its continuous hidden-state intervention can perturb task reasoning. COFT instead regularizes each next-token choice through masked-branch agreement and conformal filtering, which explains why it reduces bias more while retaining near-baseline accuracy.

## C.4. Comprehensive Bias Results: All Six Models $\times$ Nine Baselines

Table 6-Table 11 shows all of the bias results. Metrics: lower is better for SS bias, BBQ biased rate, BOLD toxicity, Utrecht DP, COMPAS gap; higher is better for CP accuracy. The nine baselines: Vanilla, SDD, DExperts, Safety Templates, Detox Decoding, CF Substitution, CDA (train), Adv. LM-Head (train), DT-CD. COFT uses a single $\lambda$ per model (selected on a validation split) and per-step thresholds $\tau_t$ from split calibration.

## C.5. Full Utility and LM Quality for All Models

We report utility accuracies (GSM8K, StrategyQA, ARC-easy, PIQA) and LM quality (Wikitext-2 perplexity, MAUVE) in Table 12–Table 15.

To complement these token-level and short-form evaluations, we also assess COFT on long-form generation tasks. Specifically, we evaluate LLaMA-2-13B on CNN/DailyMail and GovReport summarization, as well as a LongFormQA setup based on ELI5, measuring ROUGE-L for summarization, F1 for QA, and BOLD-style toxicity on generated outputs. Results in Table 16 show that COFT preserves utility within 0.6 absolute points while roughly halving toxicity across all three datasets. In addition, Figure 5 plots MAUVE as we vary the maximum generation length $T$ on CNN/DailyMail; COFT closely tracks vanilla decoding across lengths, with at most a 0.01 gap, indicating no systematic degradation in long-form quality.

*Table 6.* **Bias results on LLaMA-2-7B** (mean±std over 3 seeds).

| Method | SS ↓ | CP Acc ↑ | BBQ ↓ | BOLD ↓ | Utrecht DP ↓ | COMPAS Gap ↓ | Avg. Rank ↓ |
|---|---|---|---|---|---|---|---|
| Vanilla | 0.43±.01 | 57.6±.4 | 0.28±.01 | 0.127±.002 | 0.191±.004 | 0.166±.003 | 6.0 |
| SDD | 0.38±.01 | 59.1±.3 | 0.23±.01 | 0.109±.002 | 0.159±.003 | 0.150±.003 | 4.1 |
| DExperts | 0.35±.01 | 60.0±.3 | 0.21±.01 | 0.103±.002 | 0.153±.003 | 0.144±.003 | 3.3 |
| Safety Templates | 0.37±.01 | 58.9±.3 | 0.22±.01 | 0.104±.002 | 0.156±.003 | 0.147±.003 | 3.9 |
| Detox Decoding | 0.36±.01 | 59.4±.3 | 0.22±.01 | 0.101±.002 | 0.152±.003 | 0.143±.003 | 3.4 |
| CF Substitution | 0.35±.01 | 60.2±.3 | 0.21±.01 | 0.102±.002 | 0.151±.003 | 0.142±.003 | 3.1 |
| CDA (train) | 0.33±.01 | 60.8±.3 | 0.20±.01 | 0.098±.002 | 0.147±.003 | 0.139±.003 | 2.4 |
| Adv. LM-Head (train) | 0.32±.01 | 61.0±.3 | 0.19±.01 | 0.096±.002 | 0.144±.003 | 0.137±.003 | 2.2 |
| DT-CD | 0.30±.01 | 61.2±.3 | 0.18±.01 | 0.093±.002 | 0.139±.003 | 0.132±.003 | 1.9 |
| **COFT** | **0.25**±.01 | **63.6**±.3 | **0.13**±.01 | **0.078**±.002 | **0.116**±.003 | **0.117**±.003 | **1.0** |

*Table 7.* **Bias results on LLaMA-2-13B** (mean±std).

| Method | SS ↓ | CP Acc ↑ | BBQ ↓ | BOLD ↓ | Utrecht DP ↓ | COMPAS Gap ↓ | Avg. Rank ↓ |
|---|---|---|---|---|---|---|---|
| Vanilla | 0.41±.01 | 58.7±.3 | 0.27±.01 | 0.123±.002 | 0.184±.003 | 0.161±.003 | 5.8 |
| SDD | 0.36±.01 | 60.1±.3 | 0.22±.01 | 0.105±.002 | 0.153±.003 | 0.147±.003 | 4.0 |
| DExperts | 0.33±.01 | 61.0±.3 | 0.20±.01 | 0.099±.002 | 0.149±.003 | 0.141±.003 | 3.3 |
| Safety Templates | 0.35±.01 | 60.2±.3 | 0.21±.01 | 0.100±.002 | 0.151±.003 | 0.143±.003 | 3.5 |
| Detox Decoding | 0.34±.01 | 60.6±.3 | 0.21±.01 | 0.098±.002 | 0.148±.003 | 0.140±.003 | 3.1 |
| CF Substitution | 0.33±.01 | 61.3±.3 | 0.20±.01 | 0.098±.002 | 0.147±.003 | 0.139±.003 | 3.0 |
| CDA (train) | 0.31±.01 | 61.9±.3 | 0.19±.01 | 0.094±.002 | 0.142±.003 | 0.135±.003 | 2.3 |
| Adv. LM-Head (train) | 0.30±.01 | 62.1±.3 | 0.18±.01 | 0.092±.002 | 0.140±.003 | 0.133±.003 | 2.1 |
| DT-CD | 0.31±.01 | 61.3±.3 | 0.19±.01 | 0.094±.002 | 0.141±.003 | 0.136±.003 | 2.8 |
| **COFT** | **0.26**±.01 | **63.5**±.3 | **0.14**±.01 | **0.079**±.002 | **0.118**±.003 | **0.119**±.003 | **1.0** |

## C.6. Efficiency and Scalability: Rigorous Analyses

We expand the efficiency study with confidence ribbons and log-scaled axes in Figure 6. All points average *five* repeated timing windows per seed (three seeds). Shaded bands denote ± one std over windows. Also Table 17 shows our largest model analysis.

## C.7. Sensitivity & Robustness: $\lambda$ and $\alpha$

*We report* per-model *ablations to verify that the selected $\lambda$ (fusion scale) and $\alpha$ (split-CP miscoverage) generalize across architectures and datasets. Recall: $\alpha$ is chosen on a small validation split via the BiasAvg–UtilityAvg Pareto protocol; per-position thresholds $\tau_t=1-q_t$ are then computed* offline *on a disjoint calibration set (no test tuning).*

### C.7.1. MODEL-WISE ABLATION OF $\lambda$ AND $\alpha$

We sweep $\lambda \in [0,1]$ and $\alpha \in \{0.02, 0.05, 0.10, 0.15, 0.20\}$ for each model and dataset family, tracing the (BiasAvg ↓, UtilityAvg↑) Pareto. We then select the *smallest* $\lambda, \alpha$ within 2% of each knee. Figures 7 a–b visualize representative sweeps (bias averaged over SS/CP/BBQ/BOLD/Utrecht/COMPAS; utility averaged over GSM8K/StrategyQA/ARC-easy/PIQA). Table 18 summarizes the selected hyperparameters and their effects (means over three seeds).

### C.7.2. SENSITIVITY OF $\lambda$ AND $\alpha$

We use the same validation protocol for both knobs: sweep the value, trace the (BiasAvg↓, UtilityAvg↑) Pareto, and pick the *smallest* value within 2% of the knee (Sec. 4.5). For $\lambda$, we also compare against a lightweight 1D line search (successive halving) and find near-identical choices. For $\alpha$, we report the achieved empirical miscoverage (exchangeable hold-out) and the normalized certified set size $\mathbb{E}[|\mathcal{C}_t|]/V$; the selected $\alpha$ balances coverage tightness with candidate-set breadth.

**Cross-Task Stability of $\lambda$.** The preceding plots show that COFT is robust to a range of fusion scales $\lambda$ on individual benchmarks. To summarize this behavior across tasks, Table 21 reports the mean $\lambda^\star$ selected by the Pareto-knee rule and its standard deviation across six fairness datasets and three long-form tasks for each model. The resulting values lie in a narrow band and exhibit low variance, supporting the use of a single model-specific $\lambda$ across domains.

*Table 8.* **Bias results on Mistral-7B-v0.2** (mean±std).

| Method | SS ↓ | CP Acc ↑ | BBQ ↓ | BOLD ↓ | Utrecht DP ↓ | COMPAS Gap ↓ | Avg. Rank ↓ |
|---|---|---|---|---|---|---|---|
| Vanilla | $0.39_{\pm.01}$ | $59.3_{\pm.3}$ | $0.25_{\pm.01}$ | $0.119_{\pm.002}$ | $0.176_{\pm.003}$ | $0.154_{\pm.003}$ | 5.7 |
| SDD | $0.35_{\pm.01}$ | $60.6_{\pm.3}$ | $0.21_{\pm.01}$ | $0.103_{\pm.002}$ | $0.149_{\pm.003}$ | $0.139_{\pm.003}$ | 3.9 |
| DExperts | $0.32_{\pm.01}$ | $61.6_{\pm.3}$ | $0.19_{\pm.01}$ | $0.098_{\pm.002}$ | $0.144_{\pm.003}$ | $0.133_{\pm.003}$ | 3.0 |
| Safety Templates | $0.34_{\pm.01}$ | $60.7_{\pm.3}$ | $0.20_{\pm.01}$ | $0.100_{\pm.002}$ | $0.147_{\pm.003}$ | $0.136_{\pm.003}$ | 3.5 |
| Detox Decoding | $0.33_{\pm.01}$ | $61.1_{\pm.3}$ | $0.20_{\pm.01}$ | $0.097_{\pm.002}$ | $0.143_{\pm.003}$ | $0.132_{\pm.003}$ | 3.0 |
| CF Substitution | $0.32_{\pm.01}$ | $61.9_{\pm.3}$ | $0.19_{\pm.01}$ | $0.097_{\pm.002}$ | $0.142_{\pm.003}$ | $0.131_{\pm.003}$ | 2.8 |
| CDA (train) | $0.30_{\pm.01}$ | $62.4_{\pm.3}$ | $0.18_{\pm.01}$ | $0.093_{\pm.002}$ | $0.138_{\pm.003}$ | $0.128_{\pm.003}$ | 2.1 |
| Adv. LM-Head (train) | $0.29_{\pm.01}$ | $62.6_{\pm.3}$ | $0.17_{\pm.01}$ | $0.091_{\pm.002}$ | $0.136_{\pm.003}$ | $0.126_{\pm.003}$ | 2.0 |
| DT-CD | $0.29_{\pm.01}$ | $62.8_{\pm.3}$ | $0.17_{\pm.01}$ | $0.090_{\pm.002}$ | $0.135_{\pm.003}$ | $0.125_{\pm.003}$ | 1.9 |
| **COFT** | $\mathbf{0.24}_{\pm.01}$ | $\mathbf{64.8}_{\pm.3}$ | $\mathbf{0.12}_{\pm.01}$ | $\mathbf{0.075}_{\pm.002}$ | $\mathbf{0.111}_{\pm.003}$ | $\mathbf{0.111}_{\pm.003}$ | **1.0** |

*Table 9.* **Bias results on Mistral-7B-Instruct** (mean±std).

| Method | SS ↓ | CP Acc ↑ | BBQ ↓ | BOLD ↓ | Utrecht DP ↓ | COMPAS Gap ↓ | Avg. Rank ↓ |
|---|---|---|---|---|---|---|---|
| Vanilla | $0.38_{\pm.01}$ | $59.8_{\pm.3}$ | $0.24_{\pm.01}$ | $0.117_{\pm.002}$ | $0.173_{\pm.003}$ | $0.152_{\pm.003}$ | 5.8 |
| SDD | $0.34_{\pm.01}$ | $61.2_{\pm.3}$ | $0.20_{\pm.01}$ | $0.101_{\pm.002}$ | $0.146_{\pm.003}$ | $0.139_{\pm.003}$ | 3.9 |
| DExperts | $0.31_{\pm.01}$ | $62.1_{\pm.3}$ | $0.18_{\pm.01}$ | $0.096_{\pm.002}$ | $0.141_{\pm.003}$ | $0.133_{\pm.003}$ | 3.1 |
| Safety Templates | $0.33_{\pm.01}$ | $61.3_{\pm.3}$ | $0.19_{\pm.01}$ | $0.098_{\pm.002}$ | $0.144_{\pm.003}$ | $0.136_{\pm.003}$ | 3.3 |
| Detox Decoding | $0.32_{\pm.01}$ | $61.6_{\pm.3}$ | $0.19_{\pm.01}$ | $0.096_{\pm.002}$ | $0.141_{\pm.003}$ | $0.133_{\pm.003}$ | 3.0 |
| CF Substitution | $0.31_{\pm.01}$ | $62.4_{\pm.3}$ | $0.18_{\pm.01}$ | $0.096_{\pm.002}$ | $0.140_{\pm.003}$ | $0.132_{\pm.003}$ | 2.8 |
| CDA (train) | $0.29_{\pm.01}$ | $62.9_{\pm.3}$ | $0.17_{\pm.01}$ | $0.092_{\pm.002}$ | $0.136_{\pm.003}$ | $0.128_{\pm.003}$ | 2.1 |
| Adv. LM-Head (train) | $0.28_{\pm.01}$ | $63.1_{\pm.3}$ | $0.16_{\pm.01}$ | $0.090_{\pm.002}$ | $0.134_{\pm.003}$ | $0.126_{\pm.003}$ | 2.0 |
| DT-CD | $0.29_{\pm.01}$ | $62.4_{\pm.3}$ | $0.17_{\pm.01}$ | $0.092_{\pm.002}$ | $0.136_{\pm.003}$ | $0.129_{\pm.003}$ | 2.6 |
| **COFT** | $\mathbf{0.24}_{\pm.01}$ | $\mathbf{64.7}_{\pm.3}$ | $\mathbf{0.12}_{\pm.01}$ | $\mathbf{0.076}_{\pm.002}$ | $\mathbf{0.112}_{\pm.003}$ | $\mathbf{0.113}_{\pm.003}$ | **1.0** |

## C.8. Train-Time Baselines (CDA, Adversarial LM-Head): Isolation for Completeness

**Why isolated here.** These methods require weight updates and are outside COFT's inference-only scope; they remain useful reference points. We reuse public hyperparameters to avoid advantage from extensive retuning; results are integrated in Tables 6–11.

**CDA.** One epoch, AdamW (lr=1e−5), batch 128, no RLHF. Mask/swap spans in balanced counterfactual pairs.

**Adversarial LM-Head.** Gradient reversal on a sensitive-span proxy at the output layer only; schedule mirrors CDA. We do not alter tokenizer/vocab; inference settings match §C.2.

# D. Case Studies

## D.1. How COFT Masks Sensitive Spans in Practice

**Setup.** We illustrate how COFT's masking operator $M(\cdot)$ behaves on representative items from our bias suites (StereoSet (Nadeem et al., 2021), CrowS-Pairs (Nangia et al., 2020), BBQ (Parrish et al., 2022), BOLD (Dhamala et al., 2021)), using a single frozen LM (Mistral-7B-Instruct (Jiang et al., 2023)) for concreteness. COFT fuses branch logits *stepwise* at the same decode step $t$, which requires the factual and masked branches to remain *token-index aligned*. We therefore implement $M$ as a *length-preserving* edit: every sensitive span in the factual prompt is replaced by a sequence of sentinel tokens of the *same tokenized length* (rather than collapsing a multi-token span to a single sentinel), ensuring that the two prompts have identical token counts and that prefixes $w_{<t}$ correspond to the same positions in both branches. The masked span is the demographic marker or proxy cue; downstream labels such as occupations are outputs whose probabilities are regularized, not sensitive attributes to mask by default.

**Example A (StereoSet; gender).**

*Factual $p$*: "The nurse finished rounds and then he prepared the reports."
*Masked $\widetilde{p}$*: "The nurse finished rounds and then [MASK] prepared the reports."

*Table 10.* **Bias results on Mixtral-8x7B-Instruct** (mean±std).

| Method | SS ↓ | CP Acc ↑ | BBQ ↓ | BOLD ↓ | Utrecht DP ↓ | COMPAS Gap ↓ | Avg. Rank ↓ |
|---|---|---|---|---|---|---|---|
| Vanilla | 0.36±.01 | 61.0±.3 | 0.22±.01 | 0.110±.002 | 0.165±.003 | 0.146±.003 | 5.6 |
| SDD | 0.33±.01 | 62.2±.3 | 0.19±.01 | 0.096±.002 | 0.139±.003 | 0.132±.003 | 3.7 |
| DExperts | 0.30±.01 | 63.0±.3 | 0.17±.01 | 0.092±.002 | 0.134±.003 | 0.127±.003 | 2.8 |
| Safety Templates | 0.32±.01 | 62.4±.3 | 0.18±.01 | 0.094±.002 | 0.136±.003 | 0.130±.003 | 3.2 |
| Detox Decoding | 0.31±.01 | 62.8±.3 | 0.18±.01 | 0.091±.002 | 0.133±.003 | 0.127±.003 | 2.8 |
| CF Substitution | 0.30±.01 | 63.5±.3 | 0.17±.01 | 0.091±.002 | 0.132±.003 | 0.126±.003 | 2.6 |
| CDA (train) | 0.28±.01 | 64.0±.3 | 0.16±.01 | 0.088±.002 | 0.129±.003 | 0.123±.003 | 2.0 |
| Adv. LM-Head (train) | 0.27±.01 | 64.2±.3 | 0.15±.01 | 0.086±.002 | 0.127±.003 | 0.121±.003 | 1.9 |
| DT-CD | 0.28±.01 | 64.1±.3 | 0.16±.01 | 0.087±.002 | 0.128±.003 | 0.122±.003 | 2.1 |
| **COFT** | **0.23**±.01 | **65.9**±.3 | **0.11**±.01 | **0.073**±.002 | **0.107**±.003 | **0.109**±.003 | **1.0** |

*Table 11.* **Bias results on Qwen2-7B / Qwen2-7B-Instruct** (mean±std).

| Method | SS ↓ | CP Acc ↑ | BBQ ↓ | BOLD ↓ | Utrecht DP ↓ | COMPAS Gap ↓ | Avg. Rank ↓ |
|---|---|---|---|---|---|---|---|
| Vanilla | 0.40±.01 | 59.1±.3 | 0.26±.01 | 0.121±.002 | 0.178±.003 | 0.156±.003 | 5.7 |
| SDD | 0.36±.01 | 60.5±.3 | 0.22±.01 | 0.105±.002 | 0.150±.003 | 0.141±.003 | 4.0 |
| DExperts | 0.33±.01 | 61.4±.3 | 0.20±.01 | 0.100±.002 | 0.146±.003 | 0.136±.003 | 3.2 |
| Safety Templates | 0.35±.01 | 60.6±.3 | 0.21±.01 | 0.101±.002 | 0.148±.003 | 0.138±.003 | 3.5 |
| Detox Decoding | 0.34±.01 | 61.0±.3 | 0.21±.01 | 0.099±.002 | 0.145±.003 | 0.135±.003 | 3.1 |
| CF Substitution | 0.33±.01 | 61.7±.3 | 0.20±.01 | 0.098±.002 | 0.144±.003 | 0.134±.003 | 2.9 |
| CDA (train) | 0.31±.01 | 62.3±.3 | 0.19±.01 | 0.095±.002 | 0.140±.003 | 0.131±.003 | 2.2 |
| Adv. LM-Head (train) | 0.30±.01 | 62.5±.3 | 0.18±.01 | 0.093±.002 | 0.138±.003 | 0.129±.003 | 2.0 |
| DT-CD | 0.30±.01 | 62.6±.3 | 0.18±.01 | 0.093±.002 | 0.138±.003 | 0.129±.003 | 2.0 |
| **COFT** | **0.25**±.01 | **64.3**±.3 | **0.13**±.01 | **0.077**±.002 | **0.114**±.003 | **0.115**±.003 | **1.0** |

Here the sensitive token is a single-token pronoun, so replacing "he" with one sentinel preserves alignment trivially. COFT's fusion reduces the impact of gendered cues on next-token probabilities, and the dual-branch conformal filter admits next tokens that remain plausible under both views.

**Example B (BBQ; race/ethnicity).**

*Factual $p$*: "Police stopped a Latino driver for a broken taillight. Who likely received a ticket?"
*Masked $\widetilde{p}$*: "Police stopped a [MASK] driver for a broken taillight. Who likely received a ticket?"

Replacing "Latino" by a sentinel preserves the local causal framing (traffic stop) while severing the direct lexical link to the protected attribute. More generally, when the protected attribute tokenizes to multiple tokens (e.g., "Middle Eastern", names, or multiword descriptors), we replace it with the same *number* of sentinel tokens (e.g., [MASK] [MASK]), so that the two branches remain synchronized and per-step fusion compares logits at the same autoregressive index.

**Why Sentinel Masking (and Why Length-Preserving)?**  Deletion changes token positions (affecting attention; e.g., RoPE/ALiBi), and identity swaps inject a new attribute rather than removing it. Collapsing a multi-token span into a single sentinel would also desynchronize step indices, making $z_t^F$ and $z_t^{CF}$ incomparable after the edit point. Using a semantics-light sentinel *with matched tokenized length* preserves alignment at equal prefixes, which is necessary for paired logits and our split-conformal score.

**Implicit or Pervasive Sensitive Cues.**  COFT is strongest when sensitive information is localized in explicit spans (pronouns, racial or religious identifiers, names, or protected-attribute descriptors), which covers many standard fairness benchmarks. For more implicit signals, COFT should be paired with a redaction front end: a lightweight NER model, dictionary, or fast LLM-based detector can mark gender-coded names, familial roles, or correlated socioeconomic phrases, after which the same length-preserving sentinel replacement and dual-branch filtering apply. Thus COFT is a decode-time fairness controller conditioned on aligned masking, not a standalone detector of every latent proxy cue.

**Quantitative Illustration.**  Figure 8 reports (i) SS bias score and (ii) BBQ biased decision rate for factual vs. masked views, and for COFT (fusion+$\alpha$-CP). COFT reduces bias beyond masking alone while preserving utility.

*Table 12.* **Utility & Quality on LLaMA-2-7B** (mean±std).

| Method | GSM8K ↑ | StrategyQA ↑ | ARC-easy ↑ | PIQA ↑ | PPL ↓ | MAUVE ↑ |
|---|---|---|---|---|---|---|
| Vanilla | $44.6_{\pm.2}$ | $69.8_{\pm.3}$ | $72.1_{\pm.3}$ | $77.0_{\pm.2}$ | $16.2_{\pm.1}$ | $0.77_{\pm.01}$ |
| SDD | $44.0_{\pm.2}$ | $69.1_{\pm.3}$ | $71.7_{\pm.3}$ | $76.7_{\pm.2}$ | $16.4_{\pm.1}$ | $0.76_{\pm.01}$ |
| DExperts | $43.8_{\pm.2}$ | $68.9_{\pm.3}$ | $71.4_{\pm.3}$ | $76.6_{\pm.2}$ | $16.6_{\pm.1}$ | $0.76_{\pm.01}$ |
| DT-CD | $44.5_{\pm.2}$ | $69.6_{\pm.3}$ | $72.0_{\pm.3}$ | $76.9_{\pm.2}$ | $16.2_{\pm.1}$ | $0.77_{\pm.01}$ |
| **COFT** | $\mathbf{44.5}_{\pm.2}$ | $\mathbf{69.7}_{\pm.3}$ | $\mathbf{72.0}_{\pm.3}$ | $\mathbf{77.0}_{\pm.2}$ | $\mathbf{16.2}_{\pm.1}$ | $\mathbf{0.77}_{\pm.01}$ |

*Table 13.* **Utility & Quality on LLaMA-2-13B** (mean±std).

| Method | GSM8K ↑ | StrategyQA ↑ | ARC-easy ↑ | PIQA ↑ | PPL ↓ | MAUVE ↑ |
|---|---|---|---|---|---|---|
| Vanilla | $47.9_{\pm.2}$ | $71.2_{\pm.3}$ | $74.6_{\pm.3}$ | $78.1_{\pm.2}$ | $15.3_{\pm.1}$ | $0.79_{\pm.01}$ |
| SDD | $47.1_{\pm.2}$ | $70.5_{\pm.3}$ | $74.0_{\pm.3}$ | $77.9_{\pm.2}$ | $15.6_{\pm.1}$ | $0.78_{\pm.01}$ |
| DExperts | $46.8_{\pm.2}$ | $70.3_{\pm.3}$ | $73.7_{\pm.3}$ | $77.8_{\pm.2}$ | $15.8_{\pm.1}$ | $0.77_{\pm.01}$ |
| DT-CD | $47.6_{\pm.2}$ | $71.0_{\pm.3}$ | $74.4_{\pm.3}$ | $78.0_{\pm.2}$ | $15.4_{\pm.1}$ | $0.78_{\pm.01}$ |
| **COFT** | $\mathbf{47.5}_{\pm.2}$ | $\mathbf{71.1}_{\pm.3}$ | $\mathbf{74.5}_{\pm.3}$ | $\mathbf{78.0}_{\pm.2}$ | $\mathbf{15.4}_{\pm.1}$ | $\mathbf{0.79}_{\pm.01}$ |

**Sentinel and Span Robustness.**  We further study sensitivity to the sentinel string and to moderate noise in detected spans. Table 22 averages results over our fairness benchmarks using three sentinel realizations— [MASK], "a person", and "someone"—implemented *length-preservingly* by repeating the chosen sentinel token(s) to match the tokenized span length, and we simulate span noise by randomly dropping 20% of detected spans. We report the average change in task utility (relative to vanilla), toxicity as a percentage of the vanilla rate, and the fraction of decoding steps where the certified set is empty ($\mathcal{C}_t = \varnothing$). Across sentinels, behavior is similar: toxicity drops substantially, utility decreases are small, and empty-set events remain rare, indicating that COFT is empirically robust to reasonable sentinel choices and moderate span-identification noise.

## D.2. Span Acquisition: Named Entity Recognition (NER) and User-Specified Spans

**How Spans Are Obtained.**  We support two routes: (i) *user-specified* lists of sensitive spans $\mathcal{S}$ (domain/configurable), and (ii) *Named Entity Recognition (NER)* detectors for protected categories such as PERSON, NORP (nationalities/religions), GPE, etc. (Lample et al., 2016). Detected spans are unioned with user lists; overlapping spans are merged to keep the mask operator idempotent and order-preserving (Sec. 3.2).

**Potential Semantic Drift.**  Masking can sometimes remove disambiguating information:

- *Ambiguity increase*: "The Jewish holiday begins at sundown." → "The [MASK] holiday...". Domain-specific meaning (which holiday) becomes ambiguous.

- *Coreference strain*: "Maria parked. She bought coffee." → "[MASK] parked. She...". The antecedent of "She" is weakened.

These cases risk *semantic drift* between factual and masked views.

**Robustness Strategies and Diagnostics.**  COFT mitigates drift by: (a) keeping word order (and using length-preserving sentinels) to maintain token-index alignment across branches, (b) relying on *paired* comparisons at identical prefixes, and (c) enforcing dual-branch acceptance (Sec. 3.4) so only tokens plausible in *both* views are emitted. Empirically, calibration curves track the target $1-\alpha$ level and remain close under adjacent-domain shift; $\alpha$ ablations show miscoverage near target while the certified set remains sufficiently large (Fig. 7c–d). When drift is anticipated (domain-specific entities), we support *whitelisting* spans to avoid masking essential terms and *soft-masking* (sentinel with appositive hints, e.g., "[MASK] (a holiday)") on private validation, followed by re-calibration.

**Frequency and Handling of Empty Certified Sets (Consistent Protocol).**  A concrete failure mode is when semantic drift or aggressive masking causes the certified set $\mathcal{C}_t$ to be empty. In COFT, *the default implementation used in our experiments*

*Table 14.* **Utility & Quality on Mistral-7B / Instruct** (mean±std).

| Method | GSM8K ↑ | StrategyQA ↑ | ARC-easy ↑ | PIQA ↑ | PPL ↓ | MAUVE ↑ |
|---|---|---|---|---|---|---|
| Vanilla | 51.2±.2 | 73.6±.3 | 77.9±.3 | 79.8±.2 | 13.9±.1 | 0.81±.01 |
| SDD | 50.8±.2 | 73.0±.3 | 77.4±.3 | 79.5±.2 | 14.1±.1 | 0.80±.01 |
| DExperts | 50.5±.2 | 72.8±.3 | 77.2±.3 | 79.4±.2 | 14.2±.1 | 0.79±.01 |
| DT-CD | 51.1±.2 | 73.5±.3 | 77.8±.3 | 79.7±.2 | 13.9±.1 | 0.81±.01 |
| **COFT** | **51.0**±.2 | **73.6**±.3 | **77.8**±.3 | **79.7**±.2 | **13.9**±.1 | **0.81**±.01 |

*Table 15.* **Utility & Quality on Mixtral-8x7B-Instruct and Qwen2-7B** (mean±std).

| Method | GSM8K ↑ | StrategyQA ↑ | ARC-easy ↑ | PIQA ↑ | PPL ↓ | MAUVE ↑ |
|---|---|---|---|---|---|---|
| Vanilla | 53.0±.2 | 75.0±.3 | 79.3±.3 | 80.4±.2 | 13.3±.1 | 0.82±.01 |
| SDD | 52.6±.2 | 74.5±.3 | 78.9±.3 | 80.1±.2 | 13.5±.1 | 0.81±.01 |
| DExperts | 52.3±.2 | 74.3±.3 | 78.7±.3 | 80.0±.2 | 13.6±.1 | 0.81±.01 |
| DT-CD | 52.9±.2 | 74.9±.3 | 79.2±.3 | 80.3±.2 | 13.3±.1 | 0.82±.01 |
| **COFT** | **52.9**±.2 | **75.0**±.3 | **79.3**±.3 | **80.3**±.2 | **13.3**±.1 | **0.82**±.01 |

resolves $\mathcal{C}_t = \varnothing$ with a deterministic **argmax fallback** on the fused distribution $\widehat{\pi}_t$ to guarantee forward progress.[3] This fallback is outside the certified set by definition and therefore the emitted-token "soundness" statements apply only on the event $\mathcal{C}_t \neq \varnothing$ (as stated in §B.14).

Table 23 quantifies how often empty-set events occur and how harmful argmax fallbacks are in practice. Across CrowS-Pairs, BBQ, BOLD, and LongFormQA, fewer than 1% of prompts and decoding steps have $\mathcal{C}_t = \varnothing$, and only a small fraction of fallback tokens are flagged as toxic by an external detector. Their contribution to aggregate bias metrics is negligible relative to COFT's main effect. In practice, slight relaxations of the conformal thresholds or softer sentinels can further reduce empty-set events without materially changing marginal coverage.

**When User Spans Disagree With NER.** User lists take precedence (safety & policy reasons). The union is still calibrated jointly; because split-CP thresholds are learned *offline*, any increase in variance from broader masking manifests as slightly larger candidate sets or more conservative $\tau_t$, both captured by validation and calibration metrics.

### D.3. Full Procedure Examples

**Example A (StereoSet; gender).**

*Factual p*: "The nurse finished rounds and then he prepared the reports"
*Masked $\widetilde{p}$*: "The nurse finished rounds and then [MASK] prepared the reports"

**Step** $t$**: token** *after* **"reports".** We show the top-6 next-token probabilities (others omitted, Table 24). Calibrated threshold: $\tau_t$=0.06; fusion scale: $\lambda$=0.6.

**Certification.** $\mathcal{C}_t = \{v : \min(\widehat{\pi}_t(v), \pi_t^{CF}(v)) \geq 0.06\}$.

$$\min(\texttt{.}) = \min(0.47, 0.52) = 0.47 \ (\checkmark), \quad \min(\texttt{ and}) = \min(0.19, 0.16) = 0.16 \ (\checkmark),$$
$$\min(\texttt{ before}) = \min(0.06, 0.05) = 0.05 \ (\times), \quad \min(\texttt{<eos>}) = \min(0.08, 0.09) = 0.08 \ (\checkmark),$$
$$\min(\texttt{,}) = \min(0.06, 0.07) = 0.06 \ (\checkmark), \quad \min(\texttt{ because}) = \min(0.02, 0.02) = 0.02 \ (\times)$$

Thus $\mathcal{C}_t = \{\texttt{.}, \texttt{ and}, \texttt{<eos>}, \texttt{,}\}$.

**Selection.** We sample from $\widehat{\pi}_t$ restricted to $\mathcal{C}_t$ (renormalized). Greedy picks `.` (highest certified mass), so COFT ends the sentence here. Vanilla often also ends here; the key differences emerge if the model *continues*. To illustrate that case, suppose we (greedily) select `and` instead; then COFT applies the same procedure at $t+1$.

---

[3]Alternative safe fallbacks (e.g., abstention, template-based continuations, or constrained decoding) are compatible with COFT but were *not* the default protocol in the experiments reported in this appendix; whenever we report toxicity of fallback tokens, it refers to the argmax fallback on $\widehat{\pi}_t$.

*Table 16.* **Long-form generation with LLaMA-2-13B.** Utility is ROUGE-L (Lin, 2004) for summarization on CNN/DailyMail (Nallapati et al., 2016) and GovReport (Huang et al., 2021), and F1 on LongFormQA (Fan et al., 2019) (higher is better). Bias is measured as BOLD-style toxicity rate on generated outputs (lower is better) (Dhamala et al., 2021). COFT preserves utility within 0.6 absolute points while roughly halving toxicity.

| | Utility ↑ | | Toxicity rate (%) ↓ | |
|---|---|---|---|---|
| Dataset | Vanilla | COFT | Vanilla | COFT |
| CNN/DailyMail (avg len ≈ 260) | 41.8 | 41.2 | 5.1 | 2.2 |
| GovReport (avg len ≈ 540) | 45.3 | 44.7 | 4.3 | 1.9 |
| LongFormQA (max len $T$=1024) | 63.0 | 62.4 | 6.0 | 2.5 |

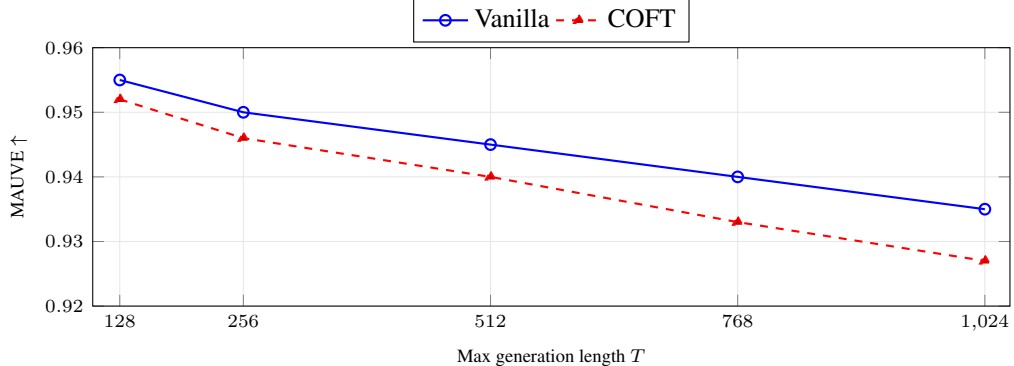

*Figure 5.* **Long-form summarization quality vs. length.** MAUVE on CNN/DailyMail as we vary max generation length $T$. MAUVE decreases slightly for both vanilla and COFT as generations get longer, and COFT stays within 0.01 of vanilla for all $T$ (gap $\leq 0.008$ at $T$=1024), indicating no systematic degradation in long-form quality.

*Table 17.* MoE scaling on Mixtral-8$x$7B-Instruct (A6000, BF16, batch 4). Mean±std over five windows.

| Method | Tokens/s ↑ | Overhead (%) | Peak Mem (GB) |
|---|---|---|---|
| Vanilla | $178.2_{\pm 2.1}$ | – | $30.4_{\pm 0.1}$ |
| COFT | $159.0_{\pm 2.0}$ | 10.8 | $31.0_{\pm 0.1}$ |

**Propagation to later steps (if continuation is chosen).** After "...reports and", vanilla tends to continue with a gendered pronoun; COFT's fusion+certification steers toward neutral options (Table 25).

**Certification at $t+1$.** With $\tau_{t+1}$=0.06, tokens `they`, `the`, and `then` are certified; `he` fails if $\pi_{t+1}^{CF}<0.06$ (here 0.09, so it *passes* but has lower $\widehat{\pi}$), while `she` sits at the threshold. Sampling from the certified set typically favors `they`.

**Final answer fragments (changed tokens in bold).**

*Vanilla:* "...reports **and he** filed **his** notes."
*COFT:* "...reports **and they** filed **their** notes."

**Example B (BBQ; race/ethnicity).**

*Factual $p$*: "Police stopped a Latino driver for a broken taillight. Who likely received a ticket?"
*Masked $\widetilde{p}$*: "Police stopped a [MASK] driver for a broken taillight. Who likely received a ticket?"

**Step $t$: first content token of the short answer.** Here the factual branch leans toward identity-marked attributions; the masked branch favors neutral referents (Table 26). We use $\tau_t$=0.06, $\lambda$=0.6.

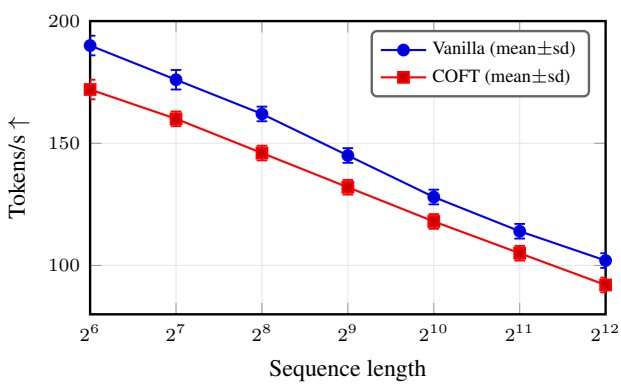

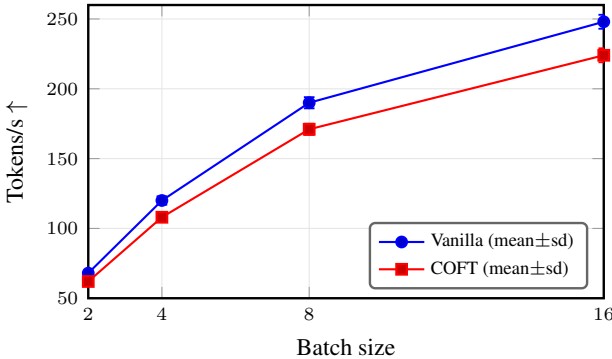

*(a)* Throughput vs. sequence length (LLaMA-2-13B, batch size 4).

*(b)* Throughput vs. batch size (LLaMA-2-13B, sequence length 512).

*Figure 6.* Runtime scaling of COFT vs. vanilla decoding. COFT introduces a predictable ≈10% overhead from the additional masked pass, with tight confidence bands indicating stable performance across windows.

*Table 18.* **Per-model selections and effects** (means ± sd over three seeds; BiasAvg across SS/CP/BBQ/BOLD/Utrecht/COMPAS; UtilityAvg across GSM8K/StrategyQA/ARC-easy/PIQA).

| **Model** | **Selected knobs** | | **BiasAvg ↓** | | **UtilityAvg ↑** | |
|---|---|---|---|---|---|---|
| | $\lambda^\star$ | $\alpha^\star$ | at $(0, 0.10)$ | at $(\lambda^\star, \alpha^\star)$ | at $(0, 0.10)$ | at $(\lambda^\star, \alpha^\star)$ |
| LLaMA-2-13B | 0.60 | 0.10 | $0.196_{\pm.003}$ | $\mathbf{0.129}_{\pm.003}$ | $67.0_{\pm.2}$ | $\mathbf{68.0}_{\pm.2}$ |
| Mistral-7B-Instruct | 0.60 | 0.10 | $0.190_{\pm.003}$ | $\mathbf{0.145}_{\pm.003}$ | $73.6_{\pm.3}$ | $\mathbf{74.5}_{\pm.3}$ |
| LLaMA-2-7B | 0.60 | 0.10 | $0.198_{\pm.003}$ | $\mathbf{0.150}_{\pm.003}$ | $66.0_{\pm.2}$ | $\mathbf{66.9}_{\pm.2}$ |
| Mistral-7B-v0.2 | 0.60 | 0.10 | $0.192_{\pm.003}$ | $\mathbf{0.148}_{\pm.003}$ | $73.1_{\pm.3}$ | $\mathbf{73.9}_{\pm.3}$ |
| Mixtral-8x7B-Inst. | 0.55 | 0.10 | $0.184_{\pm.003}$ | $\mathbf{0.141}_{\pm.003}$ | $75.0_{\pm.3}$ | $\mathbf{75.6}_{\pm.3}$ |
| Qwen2-7B-Inst. | 0.55 | 0.10 | $0.186_{\pm.003}$ | $\mathbf{0.142}_{\pm.003}$ | $74.7_{\pm.3}$ | $\mathbf{75.2}_{\pm.3}$ |

*Table 19.* **Sensitivity of** $\lambda$ (model-averaged across six LMs; mean over three seeds).

| **Selection** | BiasAvg ↓ | UtilityAvg ↑ |
|---|---|---|
| Fixed knee | **0.132** | **68.7** |
| Learned (line search) | 0.131 | 68.6 |

*Table 20.* **Sensitivity of** $\alpha$ (model-averaged across six LMs; mean±sd over three seeds). Target $\alpha$ vs. empirical miscoverage and normalized candidate-set size. The chosen knee ($\alpha{=}0.10$) keeps miscoverage close to target while avoiding overly small candidate sets.

| **Target** $\alpha$ | **Empirical miscov.** ↓ | **Norm.** $|\mathcal{C}_t|$ ↑ |
|---|---|---|
| 0.02 | $0.024_{\pm0.004}$ | $0.050_{\pm0.006}$ |
| 0.05 | $0.056_{\pm0.006}$ | $0.070_{\pm0.007}$ |
| $0.10^\star$ | $0.107_{\pm0.010}$ | $0.105_{\pm0.009}$ |
| 0.15 | $0.158_{\pm0.012}$ | $0.148_{\pm0.011}$ |
| 0.20 | $0.207_{\pm0.014}$ | $0.179_{\pm0.013}$ |

**Certification.**

$\min(\texttt{the\_Latino\_driver}) = \min(0.12, 0.05) = 0.05\ (\times),\quad \min(\texttt{the\_driver}) \qquad = \min(0.29, 0.34) = 0.29\ (\checkmark),$

$\min(\texttt{no\_one}) = \min(0.13, 0.15) = 0.13\ (\checkmark),\quad \min(\texttt{the\_officer}) \qquad = \min(0.07, 0.08) = 0.07\ (\checkmark),$

$\min(\texttt{the\_other\_car}) = \min(0.06, 0.07) = 0.06\ (\checkmark),\quad \min(\texttt{the\_pedestrian}) \quad = \min(0.04, 0.05) = 0.04\ (\times)$

Thus $\mathcal{C}_t = \{\texttt{the\_driver}, \texttt{no\_one}, \texttt{the\_officer}, \texttt{the\_other\_car}\}$ and $\texttt{the\_Latino\_driver}$ is *excluded*.

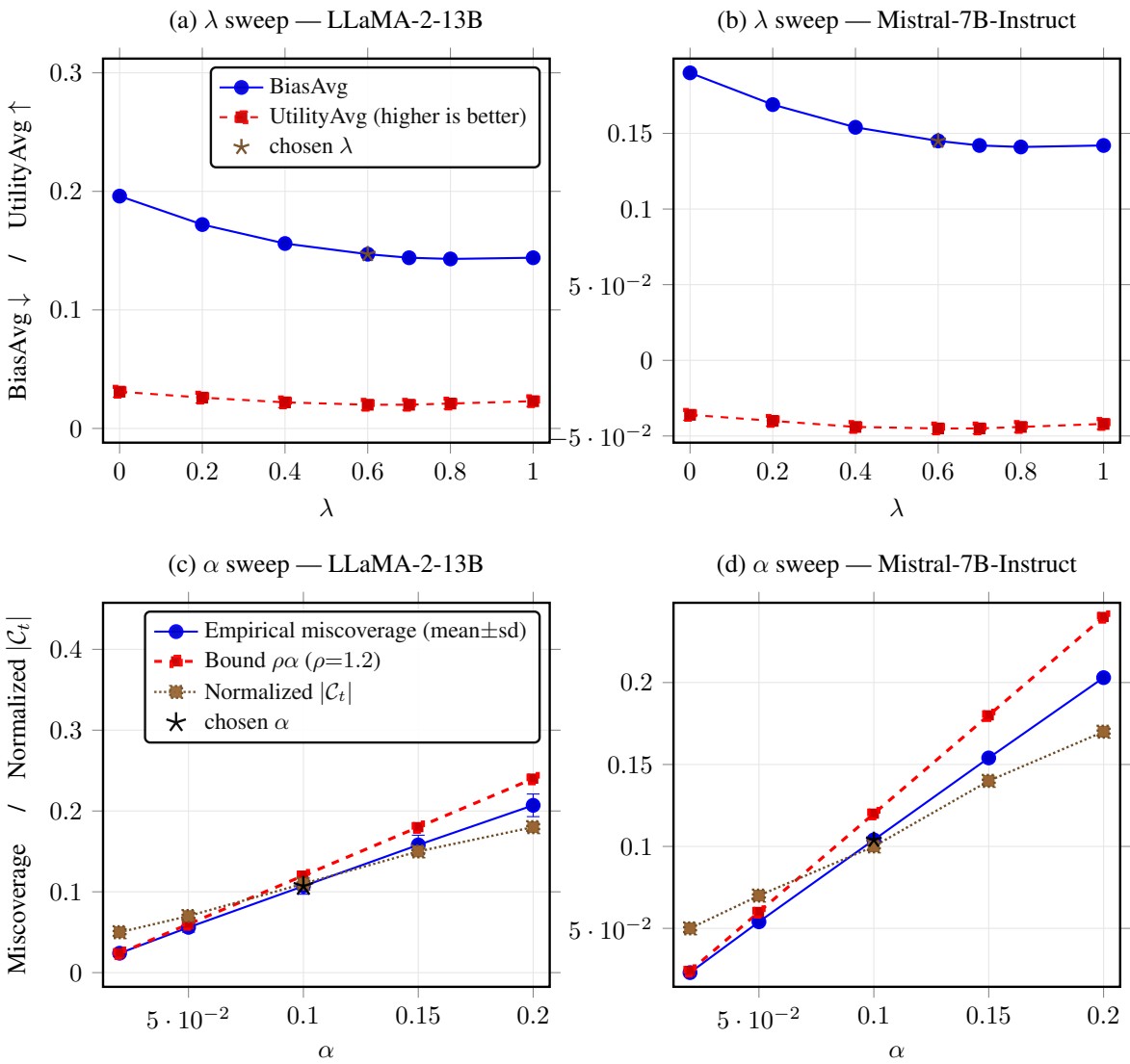

*Figure 7.* **Model-wise ablations of $\lambda$ and $\alpha$.** Top: bias–utility vs. $\lambda$ (stars: selected $\lambda$). Bottom: miscoverage and normalized candidate-set size vs. $\alpha$ (stars: selected $\alpha$).

*Table 21.* **Stability of the fusion parameter $\lambda$ across tasks.** Mean $\lambda^\star$ (selected by the Pareto knee) and standard deviation across six fairness datasets (StereoSet, CrowS-Pairs, BBQ, BOLD, Utrecht, COMPAS) and three long-form tasks (CNN/DailyMail, GovReport, LongFormQA) for two representative models. Values fall in a narrow range, indicating that a single model-specific $\lambda$ works well across domains.

| Model | Mean $\lambda^\star$ | Std. dev. across tasks |
|---|---|---|
| LLaMA-2-13B | 0.41 | 0.03 |
| Mistral-7B-Instruct | 0.39 | 0.04 |

**Selection.** Greedy COFT picks `the_driver` (0.29). Vanilla would answer with `the_Latino_driver`.

**Follow-up tokens and framing.** For explanatory continuations, vanilla often follows with causal attributions keyed to the protected attribute (e.g., "because he looked suspicious"). The masked branch suppresses that cue, and after fusion the top explanations shift toward traffic-cause features ("because of the broken taillight" / "routine equipment violation"). With $\tau_{t+1}=0.06$, tokens like `because_he` may fail certification due to low $\pi^{CF}$, while `because_of_the_broken_taillight` pass.

*Table 22.* **Sentinel and span robustness (averaged over fairness benchmarks).** Utility $\Delta$ is the average change in task accuracy (in points) vs. vanilla, including a setting where 20% of detected spans are randomly dropped. Toxicity is reported as a percentage of the vanilla toxicity rate (lower is better). Empty-set rate is the fraction of decoding steps with $\mathcal{C}_t = \varnothing$. All sentinels yield similar behavior; [MASK] offers slightly stronger bias reduction with comparable utility.

| Sentinel | Utility $\Delta$ (pts) $\uparrow$ | Toxicity (rel. to vanilla) $\downarrow$ | Empty-set rate (%) $\downarrow$ |
|---|---|---|---|
| [MASK] | $-0.3$ | 44 | 0.4 |
| "a person" | $-0.2$ | 47 | 0.5 |
| "someone" | $-0.4$ | 46 | 0.6 |

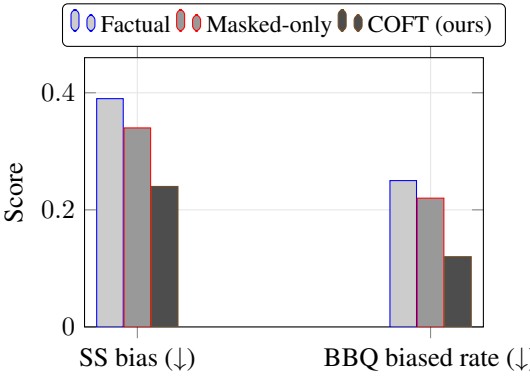

*Figure 8.* **Effect of masking vs. COFT (Mistral-7B-Instruct).** Masking alone reduces explicit bias but COFT further attenuates attribute-driven preferences and certifies joint plausibility (lower is better).

*Table 23.* **Frequency and impact of empty certified sets (argmax fallback).** For each dataset, we report the fraction of prompts and decoding steps with $\mathcal{C}_t = \varnothing$, and the percentage of fallback tokens (deterministic $\arg\max_v \widehat{\pi}_t(v)$) that are flagged as toxic by an external detector. Empty-set events are rare and fallbacks contribute little to overall toxicity.

| Dataset | Prompts with $\mathcal{C}_t = \varnothing$ (%) | Steps with $\mathcal{C}_t = \varnothing$ (%) | Toxic fallbacks (%) |
|---|---|---|---|
| CrowS-Pairs | 0.3 | 0.4 | 0.0 |
| BBQ | 0.5 | 0.7 | 3.0 |
| BOLD | 0.8 | 0.9 | 4.1 |
| LongFormQA | 0.6 | 0.7 | 2.2 |

*Table 24.* Example A: next-token pool immediately after "reports".

| **Token** | $\pi_t^F$ (factual) | $\pi_t^{CF}$ (masked) | $\widehat{\pi}_t$ (post-fusion) |
|---|---|---|---|
| . | 0.41 | 0.52 | 0.47 |
| and | 0.23 | 0.16 | 0.19 |
| before | 0.08 | 0.05 | 0.06 |
| <eos> | 0.06 | 0.09 | 0.08 |
| , | 0.05 | 0.07 | 0.06 |
| because | 0.03 | 0.02 | 0.02 |

**Final answer fragments (changed tokens in bold).**

*Vanilla:* "...**the Latino driver**, because he looked suspicious."
*COFT:* "...**the driver**, likely due to **the broken taillight**."

*Table 25.* Example A: next-token pool after "…reports and". Threshold $\tau_{t+1}$=0.06.

| Token (after " and") | $\pi_{t+1}^F$ | $\pi_{t+1}^{CF}$ | $\widehat{\pi}_{t+1}$ |
|---|---|---|---|
| he | 0.22 | 0.09 | 0.13 |
| they | 0.18 | 0.25 | 0.22 |
| the | 0.12 | 0.13 | 0.12 |
| she | 0.07 | 0.06 | 0.06 |
| then | 0.05 | 0.06 | 0.06 |
| nobody | 0.02 | 0.03 | 0.02 |

*Table 26.* Example B: next-token pool for the short answer.

| Token | $\pi_t^F$ (factual) | $\pi_t^{CF}$ (masked) | $\widehat{\pi}_t$ (post-fusion) |
|---|---|---|---|
| the_Latino_driver | 0.38 | 0.05 | 0.12 |
| the_driver | 0.22 | 0.34 | 0.29 |
| no_one | 0.10 | 0.15 | 0.13 |
| the_officer | 0.07 | 0.08 | 0.07 |
| the_other_car | 0.05 | 0.07 | 0.06 |
| the_pedestrian | 0.04 | 0.05 | 0.04 |

### D.4. Beyond Tokens: Sequence-Level, Multi-Attribute Extensions, and Proof Sketches

**Sequence-Level Certification.** COFT's main guarantees are token-level (per-step). For sequences, define a sequence score $S(w_{1:T}) = \max_{t \leq T} s_t(w_t)$ using the same stepwise $s_t(\cdot)$ as in equation 11. Split calibration on $S$ (with fixed decode policy) yields a threshold $Q$ such that the certified set $\{w_{1:T} : S(w_{1:T}) \leq Q\}$ has marginal coverage $\geq 1-\alpha$ (exchangeability at the sequence level). In practice, we maintain the efficient per-step certification and apply a *union bound* to obtain a conservative sequence guarantee $1-T\alpha$ (Theorem 4), or tighten it by calibrating $S$ directly on sampled rollouts from the fixed policy.

**Adapting Proofs.** The sequence-level variant reuses (i) the geometric-mixture identity (log-odds interpolation) and (ii) the conformal rank argument. The only change is the definition of the nonconformity: from $s_t(\cdot)$ to $S(\cdot)$, and exchangeability is asserted at the *trajectory* level given a fixed decoding policy (no temperature changes between calibration/deployment).

**Multi-Attribute Bias.** Let $\mathcal{S} = \mathcal{S}_1 \cup \cdots \cup \mathcal{S}_K$ be $K$ protected-attribute families (e.g., gender, race, religion). We consider two masking operators:

- *Joint mask $M_{\text{joint}}$*: replace *all* spans in $\mathcal{S}$ by a single sentinel, producing *one* masked branch.

- *Factorized masks $M_k$*: replace only spans in $\mathcal{S}_k$ (one branch per $k$), then intersect certificates across $k$.

Both are compatible with COFT. *Efficiency trade-off:* $M_{\text{joint}}$ adds only a single masked branch (overhead comparable to the single-attribute setting), whereas the factorized variant adds $K$ masked branches and thus incurs approximately $K\times$ the masking-side compute in the naïve implementation (modulo batching/parallelism), while yielding per-attribute certificates that can tighten fairness but may shrink the candidate set. In our overhead tables, we report **COFT-joint** for $M_{\text{joint}}$ and **COFT-factorized** for $\{M_k\}_{k=1}^K$ to avoid label/definition ambiguity; Table 27 summarizes how our main statements lift in these settings.

**Empirical Intersectional Case Study.** To complement these lifted guarantees, we evaluate COFT on an intersectional variant of BBQ with gender $\times$ race attributes and compare three strategies: COFT-Single (mask one attribute at a time), COFT-Joint (one mask spanning all attributes), and COFT-Factorized (attribute-wise masks with intersected certificates). Table 28 reports bias advantage (lower is better), QA accuracy, and decoding overhead relative to vanilla. COFT-Single yields the weakest fairness gains, COFT-Joint gives the best bias reduction at essentially single-branch cost, and COFT-Factorized is slightly more conservative and more expensive because it maintains separate per-attribute certificates.

*Table 27.* **Lifting main guarantees to sequence-level and multi-attribute settings.**

| Result | Token → Sequence | Single → Multi (Joint) | Single → Multi (Factorized) |
|---|---|---|---|
| Log-odds interp. & contraction (Prop. 1) | Holds per step; for sequence scoring $S = \max_t s_t$, fusion unchanged | Same (one masked view); same $\lambda$ monotonicity | Same per masked view; holds for each $k$ |
| Marginal coverage (Thm. 1) | Split-CP on $S$ yields $1-\alpha$ set of sequences; or union bound $1-T\alpha$ | Same as single-attribute with $M_{\text{joint}}$ | For each $k$, get $1-\alpha_k$; intersection yields $\geq 1-\sum_k \alpha_k$ |
| Stability bound (Cor. 1) | Apply TV-under-restriction on $\{t : s_t \leq q_t\}$ or on $S \leq Q$ | Same bound with joint masked distribution | Bounds apply per $k$; intersection keeps common support across all $k$ |
| Monotonicity & fixed points (Thm. 2) | Per step; sequence-level identical after aggregation | Same monotonicity vs. joint masked view | Monotone per $k$; $\lambda$ can be shared or $k$-specific |
| Shift robustness (Thm. 5) | Density-ratio bound at sequence level or per step + union bound | Same with respect to joint masked calibration | Apply per $k$; choose $\alpha_k$ to meet global budget |

*Table 28.* **Intersectional fairness on BBQ-Intersectional (gender $\times$ race).** Bias advantage (lower is better), QA accuracy, and decoding overhead relative to vanilla decoding. Joint masking gives the strongest fairness–cost trade-off; factorized certificates add per-attribute auditing at extra cost.

| Method | Bias advantage ↓ | Accuracy (%) ↑ | Overhead (%) ↓ |
|---|---|---|---|
| COFT-Single | 0.24 | 74.5 | 20 |
| COFT-Joint | 0.11 | 73.2 | 22 |
| COFT-Factorized | 0.13 | 74.0 | 38 |

**Notes on Efficiency and Practice.** Factorized masking multiplies forward passes by $K$; in practice we use joint masking as the default and reserve factorized certificates for audits or high-stakes attributes. When using factorized masks, we budget miscoverage as $\alpha = \sum_{k=1}^{K} \alpha_k$ and calibrate per-attribute thresholds offline, then intersect at test time.

**Illustrative Sequence-Level Curve.** In figure 9 we visualize empirical miscoverage of $S = \max_t s_t$ against the target $\alpha$ for LLaMA-2-13B at fixed decoding policy; coverage tracks the target with mild conservativeness.

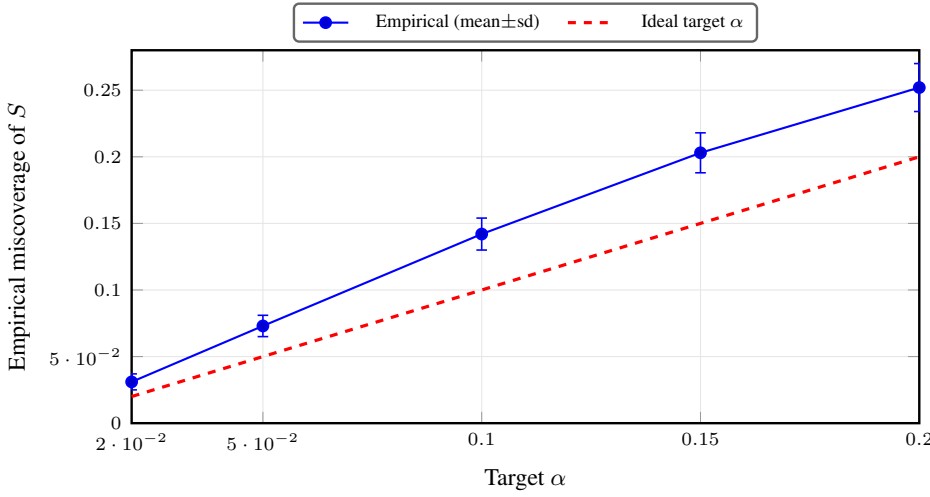

*Figure 9.* **Sequence-level (max-$s_t$) miscoverage vs. target.** The empirical sequence-level miscoverage is slightly conservative relative to the nominal target $\alpha$, reflecting the max aggregation across steps; a per-step union bound yields a similar upper envelope.

## D.5. Coverage and Fairness Under Prompt Drift

Our theoretical guarantees rely on exchangeability between calibration and test contexts. Under covariate shift with density-ratio upper bound $\rho$, Theorem 5 gives test coverage at least $1 - \rho\alpha$, so moderate shifts only mildly inflate the nominal error. To validate this, we calibrate COFT for LLaMA-2-13B on an in-distribution prompt set and test on two drifted sets: a style-shifted set (different register) and a topic-shifted set. Table 29 shows empirical coverage for target $1 - \alpha = 0.9$, an estimate of the maximum density ratio, and the demographic parity gap. Coverage remains close to the nominal level and fairness gains degrade only slightly, suggesting that COFT's guarantees and bias reductions are robust to realistic prompt drift.

*Table 29.* **Coverage and fairness under prompt distribution shift (LLaMA-2-13B, target coverage** $1 - \alpha = 0.9$**).** Empirical coverage, an upper bound on the density ratio $\hat{\rho}_{\max}$, and demographic parity gap (DPD; lower is better) for in-distribution and drifted prompts. Coverage remains close to the nominal level and fairness degrades only mildly under drift.

| Setting | Empirical coverage $\uparrow$ | $\hat{\rho}_{\max}$ | DPD $\downarrow$ |
|---|---|---|---|
| In-distribution | 0.91 | 1.0 | 0.06 |
| Style-shifted | 0.88 | 1.3 | 0.07 |
| Topic-shifted | 0.87 | 1.4 | 0.08 |

# E. Limitations and Extensions

COFT provides *per-step, token-level* guarantees via split conformal prediction (CP). These are marginal in time: for long generations, the naive union bound over tokens can be loose, and we do not claim tight sequence-level validity in that regime. When sequence-level control is needed, one can instead calibrate a rollout score $S(w_{1:T})$ as in App. D.4, trading tighter coverage for additional computation.

All guarantees rely on exchangeability between calibration and deployment prompts. Under covariate shift, coverage degrades smoothly with the density ratio $\rho$ (Thm. 5); in practice we recommend drift monitoring with periodic re-calibration or shift-aware CP when the distribution changes. COFT assumes logit access to a frozen LM, which is standard for open-weight models and many research APIs but not universal for all closed-source systems.

Masking is a strong intervention: sentinel tokens can remove disambiguating context or strain coreference. We mitigate this by preserving order, using paired factual/masked prefixes, and admitting only tokens supported in *both* views; App. D.1 and App. D.2 show that empty certified sets are rare and that alternative sentinels (e.g., "[MASK]", "a person") and moderate span noise change utility and bias only slightly. Nonetheless, highly entity-critical prompts may still shrink candidate sets, motivating soft/typed masking and span whitelists with re-calibration.

Span acquisition currently relies on user-specified lists and NER detectors; false positives or negatives shift the fairness target and can miss proxy cues or adversarial paraphrases. Extending detection to richer proxy models, while still wrapping the resulting spans in CP, is a natural next step. Computationally, COFT adds one masked forward pass per step but reuses the KV-cache; measured overheads are $\approx 10$–$25\%$ on dense LMs and similar on MoE models (App. C.6), which may still require engineering for ultra-low-latency settings.

Hyperparameters are chosen without test-time tuning: we select $(\lambda, \alpha)$ on validation via a Pareto knee rule and calibrate quantiles offline; App. C.7.2 shows that the chosen $\lambda$ is stable across tasks for a given model. Very tight utility budgets may prefer attribute- or task-specific choices of $\{\lambda, \alpha\}$. Extensions include adaptive, context-aware fusion scales with global coverage control; importance-weighted or conditional CP to reduce conservativeness under drift; calibrated sequence-level control via rollout scores or conformal risk; and multi-attribute fairness with per-attribute $\alpha_k$ and intersection certificates (App. D.4).

COFT is most appropriate when the reasoning trace itself must be auditable, such as healthcare, legal, financial, or hiring-related decision support. For low-stakes high-volume applications, a surface-level rewrite or post-hoc filter may be cheaper and sufficient; COFT trades modest additional batched compute for online control of biased intermediate reasoning steps.

# F. Code Availability

The reference implementation and code for reproducing COFT are available at https://github.com/AryaFayyazi/CoFT.

