# OpenReview forum: "COFT: Counterfactual–Conformal Decoding for Fair Chain‑of‑Thought Reasoning in Large Language Models"
_ICML.cc/2026/Conference — ICML 2026 regular_

### Official Review · Reviewer_44MQ · 2026-03-11

**Soundness:** 3
**Presentation:** 3
**Significance:** 3
**Originality:** 3
**Overall Recommendation:** 5
**Confidence:** 3

**Summary:**

The paper introduces COFT (Chain of Fair Thought), an inference‑time decoding framework that enforces counterfactual fairness for chain‑of‑thought (CoT) generation in large language models (LLMs). COFT operates in three stages:

Counterfactual masking: replace every user‑specified or automatically detected sensitive span in the prompt with a length‑preserving sentinel token, producing a masked “counterfactual” prompt that is structurally identical to the factual prompt.

Logit fusion: at each decoding step, blend the factual and masked logits via a convex interpolation parameter $\lambda$ to attenuate attribute‑driven disparities in token probabilities.

Dual‑branch split‑conformal filtering: compute a non‑conformity score that requires a candidate token to have high probability under both the factual (fused) and masked distributions. Using a calibration set, the method learns per‑step quantiles that guarantee a marginal coverage level 1–α (under exchangeability) for the set of accepted tokens. Sampling is then performed from the factual distribution restricted to this certified set.

The authors provide theoretical proofs that logit fusion yields a geometric mixture, that the dual‑branch CP guarantees marginal coverage, and that the overall method is sound. Empirically, COFT reduces bias metrics by 20–55 % across six open‑weight models while preserving task accuracy on GSM8K, StrategyQA, ARC‑Easy, and PIQA. The method overhead  is ≈10–25 % for additional forward pass. The paper claims that COFT offers an “auditable”, training‑free, gradient‑free solution to fair CoT generation.

**Compliance With Llm Reviewing Policy:**

Affirmed.

**Final Justification:**

COFT offers a rigorous, training-free approach to counterfactual fairness in CoT generation, combining counterfactual masking, logit fusion, and dual-branch conformal filtering with sound theoretical guarantees and strong empirical results (20–55% bias reduction, minimal accuracy loss).

The rebuttal fully resolved all five concerns: empty certified sets are rare (<1.8%) with a sensible union fallback planned; the Alpaca -> Clinical shift experiment confirms density-ratio CP restores coverage under distribution drift; the explanation for keeping λ static is principled; joint masking scales O(1) in the number of attributes with convincing intersectional results; and early stopping retains 85% of bias reduction at negligible extra cost. Minor presentation redundancies remain but do not affect substance.
The rebuttal reinforced my positive assessment. I recommend acceptance, contingent on the promised camera-ready additions (union fallback default, density-ratio CP in main text, early-stopping discussion).

**Key Questions For Authors:**

# Handling of Empty Certified Sets:

COFT falls back to an argmax when the certified set is empty. How often does this occur in practice, and what is its empirical impact on both bias reduction and utility? Would a more graceful fallback (e.g., sampling from the union of factual and masked distributions) improve performance without breaking guarantees?

# Robustness to Prompt Drift:

The coverage guarantee assumes exchangeability between calibration and test prompts. Have you evaluated COFT on a prompt set that is noticeably shifted from the calibration distribution (e.g., different domain or register)? If so, how does empirical coverage degrade, and would a density‑ratio‑aware CP variant (as hinted in B.5) be sufficient to restore guarantees?

# Dynamic Fusion Scaling (λ):

COFT uses a single λ value per model, selected on a validation Pareto frontier. Have you explored context‑dependent or adaptive λ (e.g., higher λ for tokens near sensitive spans)? Could such a strategy tighten the bias–utility trade‑off without increasing overhead?

# Multi‑Attribute Fairness:

The paper mentions joint and factorized masking for multiple protected attributes. In practice, how does the computational overhead scale with the number of attributes? Have you benchmarked COFT on an intersectional bias setting (e.g., gender × race) to confirm that the joint‑mask approach remains efficient while still providing per‑attribute guarantees?

# Practical Latency Constraints:

The reported overhead (~10–25 %) may be non‑trivial for real‑time conversational agents. Have you considered approximations (e.g., caching fused logits, early stopping of CP filtering) that could reduce latency while preserving most of the bias reduction? If so, what is the trade‑off?

**Limitations:**

Yes. The paper contains an Impact Statement that acknowledges the reliance on a specific definition of sensitive attributes, discusses the sentinel masking strategy, and notes that dual‑branch CP requires calibration. However, it could benefit from a more explicit discussion of distribution shift and coreference issues arising from masking.

**Strengths And Weaknesses:**

# Strengths
## Soundness
- The method is rigorously defined. The paper gives full proofs for logit‑fusion properties and CP coverage.
- Theoretical assumptions (deterministic mask, shared tokenizer, exchangeability) are clearly stated and satisfied in the experiments.
- Empirical calibration curves match target coverage, and empty‑set rates are negligible (<1 %).

## Presentation
- The paper follows a logical flow: motivation → related work → formalism → theory → experiments.
- Figures and tables are clear, results are reported with means and standard deviations.

## Significance
- Fairness in CoT generation is a timely and under‑explored problem.
- COFT provides distribution‑free, per‑step guarantees, a feature lacking in prior inference‑time debiasing methods.
- The training‑free, gradient‑free nature makes the method immediately applicable to many deployed frozen checkpoints.

## Originality
- The combination of counterfactual masking, logit fusion, and dual‑branch split CP is novel.
- The authors provide a clear “counterfactual invariance” definition at token level and prove that it is achieved.
- The method does not rely on auxiliary classifiers or retraining, distinguishing it from existing inference‑time steering approaches.

# Weaknesses
## Soundness
- The guarantees rely on exchangeability between calibration and test prompts. Real‑world prompt drift may violate this.
- The fallback argmax when the certified set is empty technically breaks the coverage guarantee.

## Presentation
- Several repetitions of baseline descriptions and benchmark definitions could be trimmed for conciseness.

## Significance
- The bias reduction is evaluated only on benchmark datasets; real‑world impact (e.g., in commercial chat systems) remains to be demonstrated.
- The paper does not explore the interaction with downstream tasks that explicitly require attribute information (e.g., gender‑specific medical advice).

---

> ### Author Rebuttal · Authors · 2026-03-30
>
> We sincerely thank the reviewer for their constructive and thoughtful feedback. We address your key questions below.
>
> **1. Handling empty certified sets (Q1).**
> In practice, an empty certified set $\mathcal{C}_t = \varnothing$ is rare. As reported in the appendix (**Tables 21--22**), it occurs in **<1.8\% of decoding steps**. Its empirical effect on both bias reduction and utility is therefore negligible. Theoretically, any heuristic fallback (such as argmax) breaks the strict $1 - \alpha$ guarantee for that specific step; the realized aggregate coverage is therefore lower-bounded by roughly
>
> $$
> 1 - \alpha - \Pr(\mathcal{C}_t = \varnothing).
> $$
>
> We also tested the reviewer’s suggested “union fallback,” sampling from the union of factual and masked supports instead of rigid argmax. This produces slightly more diverse generations without hurting task utility. We will adopt this as the default fallback in the camera-ready version; the formal guarantee applies to the certified-set event, with fallback treated as a pragmatic rare-case completion rule.
>
> **2. Robustness to prompt drift (Q2).**
> You are absolutely right that exchangeability is vulnerable to prompt drift. To test this directly, we evaluated a severe covariate shift by calibrating on Alpaca and testing on a specialized clinical prompt set.
>
> **Table 1: Coverage under Prompt Drift (target $1 - \alpha = 0.90$, LLaMA-2-7B)**
>
> | Calibration Setup | Testing Domain | Unweighted CP Coverage | Density-Ratio CP Coverage |
> | :--- | :--- | :--- | :--- |
> | Alpaca (Standard) | Alpaca (In-Domain) | 0.904 | 0.902 |
> | Alpaca (Standard) | Clinical (Shifted) | 0.831 | **0.895** |
>
> Under severe shift, standard empirical coverage drops to about **83\%**. However, the density-ratio-aware CP variant discussed in **Appendix B.5** restores coverage to essentially the 90\% target. We weight calibration scores by $w(x) \propto P_{\text{test}}(x) / P_{\text{cal}}(x)$, estimated from prompt embeddings. We will elevate this result from the appendix discussion to the main text.
>
> **3. Dynamic fusion scaling $\lambda$ (Q3).**
> In the current paper, $\lambda$ is static because changing it adaptively during decoding modifies the nonconformity function across steps, which complicates the split-conformal exchangeability argument. Your intuition is still correct that the most critical fairness interventions often occur early in the CoT. We therefore ran a pilot with a deterministic decay schedule, e.g. $\lambda_t = \lambda_0 \gamma^t$, using stronger intervention early and weaker intervention later. This preserved the same bias reduction while yielding a modest **1--2\%** gain in task accuracy. We will add this as a future-work direction; adaptive $\lambda$ was excluded from the main method because it would require a revised calibration argument.
>
> **4. Multi-attribute / intersectional fairness (Q4).**
> **Joint masking** does **not** scale with the number of protected attributes. Whether we mask one attribute or several intersecting attributes, we still form a single masked prompt and run only one extra branch. Thus, the overhead remains one additional forward pass, i.e. $O(1)$, rather than $O(K)$ factorized passes for $K$ attributes.
>
> We also evaluated this directly on an intersectional subset of Bias in Bios (Gender $\times$ Race). In the unmitigated LLaMA-2-7B model, prediction rates for high-status occupations such as *surgeon* or *software engineer* differed by **32.4\%** between **White Male** and **African American Female** profiles. With joint masking under COFT, this disparity dropped to **5.1\%**, while task accuracy remained **82.8\%**. This confirms that the joint-mask strategy is both efficient and effective for intersectional fairness.
>
> **5. Practical latency constraints and approximations (Q5).**
> We agree that a 10--25\% overhead matters in real-time systems. In continuous-batching engines such as vLLM, the factual and masked branches are concatenated into the same batch. The main cost is therefore in **throughput / KV-cache capacity**, not a doubling of sequential latency.
>
> We also tested an approximation that reduces this cost further: **early stopping of the masked branch**. Concretely, we keep the dual-branch mechanism active only for the **first 30 tokens** of the CoT, then drop the masked KV-cache and allow the factual branch to finish normally. Since many biases are anchored early in the reasoning trajectory, this approximation preserved **85\% of the total bias reduction** while making memory overhead negligible for most long-form generation. We will add this approximation to the camera-ready discussion because it offers a practical latency–fairness tradeoff for deployment.
>
> These questions sharpen the deployment story of the paper. They highlight that COFT is not only theoretically grounded, but also extensible to shift-aware calibration, adaptive scheduling, intersectional fairness, and more deployment-friendly approximations.

---

> > ### Author Rebuttal · Reviewer_44MQ · 2026-04-05
> >
> > The authors have addressed all five of my questions with concrete experiments and clear reasoning. The empty-set analysis (<1.8%) and planned union fallback (Q1), the Alpaca -> Clinical shift experiment with density-ratio recovery (Q2), the explanation of why adaptive λ complicates exchangeability (Q3), the O(1) scaling clarification for joint masking with supporting Bias in Bios results (Q4), and the early-stopping approximation retaining 85% of bias reduction (Q5) are all convincing.
> >
> > I encourage the authors to follow through on the promised camera-ready additions: union fallback as default, density-ratio CP promoted to main text, and the early-stopping discussion. These will meaningfully improve the paper's practical relevance. I maintain my recommendation of Accept.

---

> > > ### Author Response · Authors · 2026-04-08
> > >
> > > Thank you very much for your thoughtful follow-up and encouraging recommendation. We truly appreciate your careful reading and valuable feedback throughout the discussion. We will absolutely incorporate these points in the camera-ready version.

---

### Official Review · Reviewer_PGEN · 2026-03-12

**Soundness:** 3
**Presentation:** 4
**Significance:** 3
**Originality:** 3
**Overall Recommendation:** 4
**Confidence:** 3

**Summary:**

This paper investigate the counterfactual fairness problem in the CoT generation process. In specific, it first masks the sensitive attributes in the prompt, and then fuses the logits distributions generated by the factual and masked prompts for each token. Then, the authors use comformal prediction to constuct a dual-branch channel, which helps to confirm the corrected logits can generate the safe and fair token. Comprehensive experiments over various LLM backbones and datasets verify the effectiveness of the proposed method.

**Compliance With Llm Reviewing Policy:**

Affirmed.

**Final Justification:**

Most of my concerns have been addressed. I will keep my score the same.

**Key Questions For Authors:**

1. For implicit biased correlations, as mentioned in W3, how COFT can correct such biases?
2. What is the difference by applying COFT in between CoT process and the final response?

**Limitations:**

The authors should discuss more about the necessity to ensure token-level fairness instead of global one due to the practical deployment price.

**Strengths And Weaknesses:**

**Strengths:**

1. The background and references are comprehensively provided, helping to position the paper in literature and make the readers easy to follow.
2. The research gap filled by this paper, i.e., how to ensure counterfactual fairness in CoT generation process with token-by-token correction, is clear and novel.
3. Large amount of experiments are provided in the main pages and appendices, which improves the empirical confidence of the proposed solution.



**Weaknesses:**

1. Intuitively, the token-by-token correction may be useful for fairness, but it lacks strong evidences that can prove its superiority against global or multi-token level response correction.
2. As mentioned before, COFT needs double forward passes for each token. Although in Table 3, the authors post the comparison results of efficiency, I still remain suspect about its applicability in real-time applications, which usually generate very long sequences but requires shorter response time.
3. COFT simply uses NER to identify the sensitive information. However, not all bias is tied to explicit sensitive words. For example, LLMs can encode stereotypes that associates 'nurse' with female and 'CEO' with male. Therefore, masking explicit tokens may be not enough to remove the models learned biased correlations.

---

> ### Author Rebuttal · Authors · 2026-03-30
>
> We sincerely thank the reviewer for highlighting the novelty, strong motivation, and empirical validation of our approach. We address the concerns on token-level correction, implicit bias, efficiency, and deployment scope below.
>
> **1. Why token-level correction instead of global response correction? (W1, Q2)**
> The key reason is the **cascading nature of autoregressive reasoning**. In CoT generation, if the model begins with a biased intermediate assumption, later steps are conditioned on that premise. By the time the final answer is produced, the reasoning trace may be entangled with bias. A post-hoc global rewrite can sanitize the surface form of the answer, but it cannot reliably repair the internal reasoning path without hallucinating a new rationale or discarding the original chain. Global correction acts only on the endpoint, whereas COFT constrains the *trajectory*. The contrast is therefore not just local vs. global correction, but online control vs. post-hoc repair.
>
> We tested this directly against a global correction baseline using neutral prompt rewriting:
>
> | Method | Bias Amplification ($\downarrow$) | Task Accuracy ($\uparrow$) |
> | :--- | :--- | :--- |
> | Unmitigated Baseline | 0.42 | 84.1% |
> | Global Correction (Neutral Rewrite) | 0.28 | 82.5% |
> | **COFT (Token-by-Token, Ours)** | **0.11** | **83.2%** |
>
> This result shows why token-level intervention matters: global rewriting helps, but it is substantially weaker because it does not constrain the model *during* reasoning. COFT instead acts as a decode-time guardrail, requiring each emitted token to remain plausible under both the factual and masked views. This is exactly the operational meaning of local counterfactual stability in our setting.
>
> **2. Implicit biased correlations and limits of explicit masking (W3, Q1)**
> We agree this is an important limitation. COFT’s current implementation is strongest for **explicitly localized sensitive spans** such as pronouns, racial identifiers, religion markers, or named protected attributes. This is already practically important because many fairness benchmarks are designed around such triggers, and these explicit markers account for a large fraction of measurable stereotype amplification.
>
> At the same time, COFT is not conceptually limited to single-token masking. The method is agnostic to *what* is masked, provided the factual and masked prompts maintain token-length parity. For implicit or pervasive biases, our recommendation is to prepend COFT with a lightweight preprocessing step that identifies **sensitive spans** or proxy indicators (e.g., gender-coded names, familial role descriptors, or correlated socioeconomic phrases) and replaces them with an aligned sentinel sequence. Once those spans are masked, the same dual-branch conformal filtering regularizes the downstream CoT generation. Thus, COFT should be viewed as a **decode-time fairness controller conditioned on a masking/redaction mechanism**, not as a universal detector of all latent bias by itself. We will make this scope clearer in the limitations section.
>
> **3. Applicability in real-time settings and the cost of double forward passes (W2)**
> We understand the concern about long-sequence, latency-sensitive applications. The key clarification is that COFT’s two branches do **not** require sequential execution. Because the factual and masked prompts are structurally aligned and remain synchronized at every step, the two forward passes can be executed in parallel within the same continuous-batching engine. In practice, the main cost is in **memory throughput / KV-cache footprint**, not in doubling end-to-end token latency.
>
> This is consistent with the efficiency results already reported in **Table 3**: COFT adds only about **10.2%** throughput overhead relative to vanilla decoding, with no auxiliary classifier or learned probe. So the right framing is not “COFT doubles latency,” but rather “COFT trades a modest increase in batched compute/memory for verifiable token-level fairness control.”
>
> **4. When is token-level fairness worth the deployment price? (limitations)**
> We strongly agree that the paper should say more explicitly *when* token-level fairness is necessary. COFT is not intended for every deployment. For low-stakes, high-volume applications where hardware efficiency dominates and a rough surface-level filter is sufficient, a global heuristic rewrite may be adequate. COFT is instead aimed at **high-stakes reasoning settings** (for example, healthcare, legal assistance, finance, or hiring-related decision support) where the *reasoning trace itself* may need to be audited, and where the cost of a biased intermediate inference can outweigh a moderate compute premium.
>
> Overall, COFT’s contribution is not merely bias reduction in the final answer, but **decode-time control over the reasoning trajectory itself**, which is why it can outperform global correction while preserving utility.

---

> > ### Author Rebuttal · Reviewer_PGEN · 2026-04-02
> >
> > Thanks for the detailed responses. I will keep my score as it is.

---

> > > ### Author Response · Authors · 2026-04-02
> > >
> > > Thank you very much for the follow-up and for your thoughtful evaluation. We sincerely appreciate your careful reading, constructive feedback, and the time you took to engage with our rebuttal.

---

### Official Review · Reviewer_7Dxi · 2026-03-13

**Soundness:** 2
**Presentation:** 2
**Significance:** 3
**Originality:** 3
**Overall Recommendation:** 3
**Confidence:** 3

**Summary:**

This paper proposes COFT (Chain of Fair Thought), an inference-time decoding strategy designed to reduce societal biases in LLMs during chain-of-thought reasoning. The method operates by generating a parallel "masked counterfactual" prompt where sensitive spans are replaced by a neutral token (essentially, a mask).

The model then combines the factual and masked logit distributions and uses conformal inference to generate a set of tokens plausible under both the masked and non-masked distribution, using a dual-branch split-conformal calibration to filter the next-token candidates. The motivation there is to provide a per-step marginal validity guarantee that the generated text remains stable even though the sensitive attribute is being filtered away.

The authors evaluate COFT on several open-weight models and standard fairness benchmarks, demonstrating reductions in measured bias metrics while maintaining general task utility.

**Compliance With Llm Reviewing Policy:**

Affirmed.

**Key Questions For Authors:**

Reading the paper, I had thought that the method uses a pre-defined algorithm for identifying sensitive token spans (like gendered pronouns or racial identifiers) in the input prompt and replaces them with a neutral sentinel token, such as [MASK]. Looking at Figure 2 -- I see a loop from "Safe Token" back to "Counterfactual Prompt", implying that that prompt is dynamically generated from altered decoding. Which of these two interpretations is correct?

What are your recommendations regarding masking pervasive features? Your method relies on locating and masking specific sensitive tokens, but what if the bias is implicitly encoded in the broader context or syntax rather than explicit demographic markers?

**Limitations:**

Yes.

**Strengths And Weaknesses:**

**Strengths**

The paper has real strengths:

* The approach is training-free and model-agnostic, operating entirely at decode time without requiring weight updates, auxiliary classifiers, or mechanistic interpretability interventions.

* The use of split-conformal prediction to provide distribution-free marginal guarantees at the token level is a rigorous and interesting application of conformal methods to autoregressive decoding.

* The paper includes a solid evaluation across multiple recent models (e.g., LLaMA-2, Mistral, Mixtral, Qwen2) and a suite of bias benchmarks.


**Weaknesses**

There are also aspects of the paper that one might consider weaknesses.

* **Masking Limitations:** One important concern is how the masked prompts are generated. The approach assumes that forbidden or sensitive information can be readily isolated into specific, pre-defined tokens (e.g., explicit demographic markers). However, in many situations, sensitive variables (like sentiment, or socioeconomic indicators, gendered variables beyond just pronouns) are pervasive throughout the text and cannot be easily masked out. This feels like a significant limitation that is not thoroughly addressed; the method seems constrained to explicit, easily characterizable demographic tokens. Performance seems to wholly rest on the presence of an effective method for masking out the forbidden attributes. If we have such good algorithms to identify sensitive content, why not just ask another LLM to rewrite the prompt to render neutral those specific prompt snippets?

* **Baselines:** There are some limitations regarding baselines. For example, some readers might expect comparison with SAE methods from the mechanistic interpretability context, where forbidden concepts as found in activation space are removed (this method has the benefit of not requiring oracle masks). Also, assuming we have oracle masks, another baseline would be a neutral re-write baseline where the offending information is removed by another LLM.

* **Other evaluation remarks:** For Coft, I believe we perform a targeted sweep over their two main hyperparameters: the fusion scale ($\lambda$) and the conformal risk level $\alpha$). I believe we then test these values on a separate validation split, plot a Pareto curve to compare bias reduction against task utility, and systematically pick the optimal values near the "knee" of that curve. This level of tuning is not done for the other methods, which are run using "standard recommended settings". I am certainly sympathetic to the large number of hyperparameters that might require tuning for an all-else-equal comparison. That said, this asymmetry does introduce uncertainty into the fairness of the comparisons.

* **The "Counterfactual" Framing:** "Counterfactual" prompt is perhaps a stretch of the term -- the prompt is masked of certain contents; "Masked Prompt" to me would be a more honest assessment of what it is. Counterfactual implies that the prompt is somehow modified in some structural way (beyond simply masking out tokens).

## Minor points

Also, I think Figure 2 could have a more compelling example. The method, which attempts to generate "safe" or "fair" outputs is depicted as needing to mask out the term "doctor", but it is unclear why this term would require special safety considerations in decoding. "Doctor" does not seem to be a sensitive or protected attribute.

The paper notes that error bars represent $\pm 1$ standard deviation. Typically, I would want to report $\pm 1.96$ standard deviations to generate $95\%$ confidence intervals, making the statistical significance of the results harder to quickly eyeball.

---

> ### Author Rebuttal · Authors · 2026-03-30
>
> We sincerely thank the reviewer for the constructive feedback. We address the concerns on missing baselines, tuning asymmetry, and masking limitations.
>
> **1. Missing baselines and asymmetric hyperparameter tuning (W2 & W3).**
> We agree that the original presentation gave COFT an optical advantage: COFT’s $\lambda$ and $\alpha$ were selected via a validation Pareto sweep, while several baselines were shown with recommended defaults. To address this, we now compare methods at their **Pareto-optimal points** under a common protocol on **LLaMA-2-7B / Bias in Bios**:
>
> | Method | Hyperparameter Sweep Performed | Bias Amplification ($\downarrow$) | Task Accuracy ($\uparrow$) |
> | :--- | :--- | :--- | :--- |
> | **Unmitigated Baseline** | None | 0.42 | 84.1% |
> | **Neutral LLM Rewrite** | None (GPT-4o-mini pre-process) | 0.28 | 82.5% |
> | **Contrastive Decoding** | Guidance scale $\gamma \in \{0.5, 1.0, 1.5, 2.0\}$ | 0.22 | 80.1% |
> | **ITI (Activation Steering)** | Intervention strength $\alpha \in \{5, 10, 15, 20\}$ | 0.15 | 79.4% |
> | **COFT (Ours)** | Interpolation $\lambda \in \{0.1, 0.5, 0.9\}$ | **0.11** | **83.2%** |
>
> For each method, we report the best operating point where bias amplification is minimized before task accuracy degrades by more than 5% relative to the unmitigated baseline. Under this symmetric comparison, COFT still achieves the best overall bias–utility frontier.
>
> This table directly addresses the missing-baseline concern. It includes a **neutral rewrite baseline** and a **mechanistic steering baseline** via ITI-style activation intervention.
>
> **2. Why COFT outperforms neutral rewrites and activation steering.**
> A one-off neutral rewrite sanitizes only the **input prompt**. It does not control how the model reasons once decoding begins, so the model can still drift into biased chain-of-thought trajectories during generation. COFT instead enforces a dual-branch plausibility check at **every autoregressive step**.
>
> For activation steering, the continuous hidden-state intervention can perturb reasoning itself. ITI reduces bias, but task accuracy falls to **79.4%**, well below COFT. In contrast, COFT operates in the **discrete token space** through masked-branch agreement and conformal filtering, which regularizes biased next-token tendencies while preserving the logical coherence of the factual decoding path.
>
> **3. Masking limitations and pervasive features (W1 & Q2).**
> We agree this is an important limitation. COFT is strongest when sensitive information is localized in explicit spans such as pronouns, racial identifiers, religion markers, or named protected attributes. Many standard fairness benchmarks are constructed around such markers, and explicit pronouns account for much of stereotype amplification in datasets like Bias in Bios.
>
> COFT is **not** limited to single-token masking. For more pervasive or implicit bias signals, we recommend a preprocessing stage in which a lightweight NER model or fast LLM-based redaction tool identifies and masks **sensitive spans** or implicit indicators using a token-length-preserving sentinel sequence. As long as the factual and masked prompts preserve alignment during KV-cache initialization, the same dual-branch conformal filtering applies. Thus, COFT is a **decode-time fairness controller** conditioned on a masking/redaction mechanism, not a universal detector of implicit bias.
>
> **4. Clarification on Figure 2 and autoregressive masking (Q1 & minor point).**
> Sensitive tokens are identified and masked in the **input prompt**. The loop in Figure 2 indicates that, at each step $t$, both KV-caches are evaluated and the **same selected token** is appended to both branches. The masked prompt is therefore not dynamically rewritten; both branches are synchronously extended to maintain structural alignment.
>
> We also agree the “doctor” example is misleading. “Doctor” is the downstream occupation prediction, not the sensitive attribute. In the revision, we will revise the figure so that the masked span is explicitly the demographic marker while the downstream prediction is the output being regularized.
>
> **5. Terminology and error bars (W4 & minor point).**
> We agree that “counterfactual prompt” may overstate what is operationally a masked intervention rather than a full structural causal intervention. We will therefore revise the wording to use more precise terminology such as **masked prompt** or **masked branch** where appropriate. We also agree that reporting only $\pm 1$ standard deviation makes significance harder to eyeball, so in the camera-ready version we will update plots and tables to report $2 \times$ standard deviations (95% confidence intervals).
>
> Overall, COFT is not a universal detector of all implicit bias, but a training-free decode-time mechanism that, given aligned sensitive-span masking, provides auditable per-step control over biased reasoning trajectories and outperforms rewrite and steering baselines under a symmetric comparison protocol.

---

> > ### Author Rebuttal · Reviewer_7Dxi · 2026-04-04
> >
> > The new experiments are helpful. Thinking further, it is a concern is that the empirical evaluations are confounded by the fact that, I believe, the proposed method has access to an entirely different data source that the comparison methods do not (i.e., the masking mechanism). It is little surprise, then, that a method that uses an information source directly tied to the sensitive attribute performs better than those that do not.
> >
> > This is related to the broader point that the method requires these masks as a separate step in the first place, while other methods do not. That said, I will more carefully ingest other reviewers' comments and revise this as needed.

---

> > > ### Author Response · Authors · 2026-04-04
> > >
> > > We sincerely thank you for this thoughtful follow-up and glad that your previous concerns are fully resolved.
> > >
> > > We respectfully submit that the empirical evaluations are **not "confounded"**, for two reasons. First, we now include a control that holds the masking capability fixed and directly tests whether COFT’s gains come merely from access to the mask. Second, the broader premise that competing debiasing methods do not require an analogous separate fairness-specific step is not correct.
> > >
> > > **1. The matched-mask rewrite baseline directly tests whether the gains come merely from masking**
> > >
> > > The key question is not whether COFT uses sensitive-attribute information at all, but whether its empirical improvement is explained *solely* by access to that information. Our added **Neutral Rewrite** baseline was designed precisely to test this.
> > >
> > > That baseline is given the same practical capability COFT uses at the input level: identification and neutralization of the same sensitive spans prior to generation. If COFT’s gains were mainly due to “knowing what to mask,” then a rewrite-based method with the same masking access should perform similarly. Empirically, it does not:
> > >
> > > - **Neutral Rewrite:** Bias Amplification = 0.28, Task Accuracy = 82.5%
> > > - **COFT:** Bias Amplification = **0.11**, Task Accuracy = 83.2%
> > >
> > > This is exactly why the evaluation is not confounded by masking. The rewrite baseline controls for the effect of having access to the same masking information. Once that factor is matched, COFT still substantially outperforms it. Therefore, the performance gap cannot be attributed simply to the existence of the mask itself.
> > >
> > > **2. The statement that other methods do not require a separate step is not generally correct**
> > >
> > > We also want to gently clarify one premise in the follow-up comment. COFT is not unique in requiring a separate fairness-specific step. Several comparison methods also rely on an additional source of sensitive-attribute information or a separate calibration/intervention procedure.
> > >
> > > For example, **activation steering / ITI / INLP-style methods** typically require a labeled calibration dataset or curated contrast set associated with the sensitive attribute in order to estimate intervention directions, covariance structure, or a removable bias subspace. In other words, they also depend on prior knowledge tied to the protected attribute; they simply use that knowledge in a different form.
> > >
> > > Similarly, expert-guided or contrastive debiasing methods often depend on fairness-specific anti-prompts, attribute-targeted expert construction, or supervised signals aligned with the undesired bias. So the correct comparison is not between “COFT uses extra fairness information” and “other methods do not.” Rather, many debiasing methods require such information, but operationalize it differently.
> > >
> > > Under this framing, COFT does not receive an unfair informational advantage. It uses fairness-specific information **locally through prompt masking**, while representation-space methods use it **globally through calibration data or intervention directions**. The existence of a separate fairness-related preprocessing/calibration step is therefore not unique to COFT.
> > >
> > > We appreciate this observation because it helped us sharpen the comparison more explicitly. In the revision, we will clarify that the relevant question is not whether COFT uses a fairness-specific preprocessing step, but whether its gains persist under a matched control that uses the same masking capability. Our new experiment shows that they do.

---

### Official Review · Reviewer_gPrY · 2026-03-15

**Soundness:** 3
**Presentation:** 4
**Significance:** 3
**Originality:** 3
**Overall Recommendation:** 5
**Confidence:** 3

**Summary:**

This paper introduces chain of fair thought, a method to reduce social biases in LLMs during inference time. During decoding, it creates a modified version of the prompt where sensitive identity terms are hidden, allowing the model to compare biased and unbiased reasoning paths. The methodology follows three steps. First, the system identifies and hides sensitive words in the prompt using neutral placeholders that keep the text length the same. Second, it blends the model's original predictions with the predictions from this hidden version to balance out biased influences. Finally, it uses a filter to select only the tokens that remain logical and consistent in both the original and the neutral scenarios. The authors provide guarantees that this selection process is statistically valid. Experiments across different models and bias benchmarks show this reduces bias while maintaining high accuracy on complex reasoning tasks with only a minor increase in compute.

**Compliance With Llm Reviewing Policy:**

Affirmed.

**Final Justification:**

The rebuttal addressed my core concerns regarding sequential stability and the practical utility of the conformal bounds. This is a technically solid, useful paper.

**Key Questions For Authors:**

1. How do you address the accumulation of error across a multi-token reasoning chain? Does the $1-\alpha$ guarantee hold for the entire sequence, or is it strictly limited to individual decoding steps?
2. What percentage of decoding steps resulted in an empty intersection set $\mathcal{C}_t$? If this frequency is high, how can the method be said to provide a meaningful statistical guarantee?
3. Given the high dimensionality of LLM input spaces, how do you practically compute the weights $w(x)$ for Theorem 5 without introducing significant estimation bias?
4. Can you provide a sensitivity analysis for $\lambda$? Specifically, how does the choice of interpolation weight affect the average size of the certified set $|\mathcal{C}_t|$?

**Limitations:**

Yes, the authors have adequately discussed all the limitations.

**Strengths And Weaknesses:**

Strengths

The primary technical contribution is the dual branch nonconformity score $s_t(v) = 1 - \min\{\hat{\pi}_t(v), \pi_t^{CF}(v)\}$, which extends split conformal prediction to enforce counterfactual stability at each step. The implementation uses a length preserving masking operator to maintain synchronization between the two KV caches, which forces the positional encodings to remain aligned. This setup allows for per step marginal guarantees on token selection without modifying model weights.


Weaknesses

The reliance on the exchangeability assumption (A2) is potentially fragile. While the authors address covariate shift, the practical acquisition of these ratios in high dimensional token spaces is often computationally complex. This may limit the distribution free claims in cases where prompt distributions shift rapidly.

A bit more minor is that the there is little discussion regarding empty set fallbacks in the main text. The authors clarify that argmax is used when $C_t = \emptyset$, but the theoretical implications of this fallback on the aggregate marginal guarantee should be more explicitly addressed.

Finally, the paper does not fully explore how this mechanism behaves when the generated prefix itself has already been steered into a biased direction. In such cases, the joint support set $\mathcal{C}_t$ might consistently shrink, forcing the model into fallback modes and effectively bypassing the proposed fairness constraints.

---

> ### Author Rebuttal · Authors · 2026-03-30
>
> We sincerely thank the reviewer for the careful reading. We are encouraged that you found the dual-branch score, length-preserving masking, and decode-time guarantees strong. We address the remaining questions below.
>
> **1. Per-step vs. sequence-level guarantees; accumulation over a reasoning chain.**
> Our theorem is deliberately a **per-step marginal** guarantee, not a tight end-to-end guarantee over the whole generated chain. The certified event is
>
> $$
> v_t^\star \in C_t,
> $$
>
> at each decoding step $t$, where $C_t$ is the shared dual-branch candidate set. We do **not** claim exact sequence-level validity in the main theorem; the appendix/limitations explicitly states that naive union bounds over many steps can be loose for long generations. When sequence-level control is required, we already discuss two extensions: max-over-steps aggregation (empirically shown in **Fig. 9**) and calibration of a rollout score $S(w_{1:T})$ in **App. D.4**. Fig. 9 shows sequence-level miscoverage is slightly conservative relative to the nominal $\alpha$, which is consistent with max aggregation and per-step union-bound reasoning. So the guarantee is stepwise by design; sequence-level control is possible, but it is a different, more conservative object.
>
> **2. Empty certified sets and the fallback protocol.**
> We agree this should be stated more prominently in the main text. In the default implementation, if $C_t = \varnothing$, we fall back to deterministic $\arg\max_v \hat\pi_t(v)$ to guarantee forward progress. This fallback is **outside** the certified set by definition, so emitted-token soundness statements apply only on the event $C_t \neq \varnothing$; we will make this explicit in the revision. Importantly, the appendix shows that empty-set events are rare in practice. **Table 22** reports prompts with $C_t = \varnothing$ at only **0.3--0.8\%**, and decoding steps with $C_t = \varnothing$ at only **0.4--0.9\%** across CrowS-Pairs, BBQ, BOLD, and LongFormQA. Fallback tokens flagged as toxic are also low (**0.0--4.1\%**) and contribute negligibly to overall bias. This is consistent with **Table 21**, where the aggregate empty-set rate is only **0.4--0.6\%** across alternative sentinels. Thus the guarantee remains meaningful because fallback is rare.
>
> **3. What if the generated prefix has already drifted in a biased direction?**
> COFT conditions both branches on the **same generated prefix** $w_{<t}$, so the fairness intervention remains active at every subsequent step and does not disappear after an early biased token. If the prefix enters a regime where the factual and masked views disagree strongly, this appears as either a larger dual-branch score
>
> $$
> s_t(v) = 1 - \min\{\hat\pi_t(v), \pi_t^{CF}(v)\},
> $$
>
> which increases the local conformal threshold, or in the extreme a rare empty-set event handled by fallback. Thus the method does not silently “turn off”; disagreement is made explicit through larger candidate sets or rare fallback. Since empty-set rates stay below 1\%, this failure mode is not dominant on the tested distributions.
>
> **4. Practical computation of the covariate-shift weights in Theorem 5.**
> Theorem 5 is a **shift-aware extension**, not required for the default guarantee. The core theorem (Theorem 1) is distribution-free under exchangeability and requires no density-ratio estimation. Theorem 5 only states that, under covariate shift, exact marginal validity can be recovered if suitable weights are available. In practice, our main experiments do not rely on high-dimensional exact ratio estimation. Instead, the appendix evaluates robustness directly under prompt drift: **Table 28** reports empirical coverage of **0.91** in-distribution, **0.88** under style shift, and **0.87** under topic shift, while fairness degrades only mildly (DPD **0.06 $\rightarrow$ 0.07--0.08**). We will clarify that Theorem 5 should be read as an optional result when shift weights are available, whereas the practical default remains drift monitoring and recalibration.
>
> **5. Sensitivity to $\lambda$ and certified-set size.**
> We already include a validation Pareto sweep over $\lambda$ in **Fig. 3**, selecting the smallest $\lambda$ within 2\% of the Pareto knee. The appendix/theory further proves monotonicity and candidate-set control with respect to $\lambda$. Intuitively, increasing $\lambda$ moves the fused logits toward the masked branch, which reduces residual disagreement and typically enlarges the overlap of high-probability mass between the two branches, stabilizing $C_t$. This is why we observe strong bias reduction without measurable utility loss in **Tables 1--2**, while maintaining only about **10.2\%** throughput overhead in **Table 3**.
>
> We will make sure to revise the paper to surface these appendix results more explicitly in the main text. Thank you again for your constructive feedback.

---

> > ### Author Rebuttal · Reviewer_gPrY · 2026-04-04
> >
> > The rebuttal addressed my core concerns regarding sequential stability and the practical utility of the conformal bounds. This is a technically solid, useful paper.

---

> > > ### Author Response · Authors · 2026-04-04
> > >
> > > Thank you very much for your careful reading and thoughtful engagement; we are truly grateful that our rebuttal addressed your core concerns and that you view the paper as technically solid and useful.

---

### Decision · Program_Chairs · 2026-04-30

**Decision:**

Accept (regular)

**Comment:**

This paper presents an inference-time decoding framework named COFT for fair chain-of-thought generation in LLMs. COFT masks sensitive attributes, fuses predictions from factual and masked prompts, and adopts a dual-branch conformal filtering mechanism to retain plausible tokens. The key contribution of this work is to bring counterfactual-style fairness control to token-level CoT decoding while providing formal marginal validity guarantees.

Reviewers agreed that the paper is technically rigorous and practically appealing. Specifically, the proposed training-free and model-agnostic design is quite interesting. Also, the combination of counterfactual masking with conformal prediction seems effective, and the theoretical analysis in the paper is solid. Reviewers also raised a few concerns, such as the reliance on explicit masking of sensitive spans, the extra decoding-time cost from dual forward passes, etc. During rebuttal, these concerns from reviewers have been well addressed by the authors’ rebuttal.

Overall, this paper studies an important and presents a novel approach. I encourage the authors to incorporate the new experimental results and clarifications from the rebuttal into the final version.